# SymMaP: Improving Computational Efficiency in Linear Solvers through Symbolic Preconditioning

**Hong Wang[1,2,3] , Jie Wang[1,2,3]\*, Minghao Ma[1], Haoran Shao[1], Haoyang Liu[1,2,3]**

[1] University of Science and Technology of China
[2] CAS Key Laboratory of Technology in GIPAS, University of Science and Technology of China
[3] MoE Key Laboratory of Brain-inspired Intelligent Perception and Cognition, University of Science and Technology of China
wanghong1700@mail.ustc.edu.cn, jiewangx@ustc.edu.cn

## Abstract

Matrix preconditioning is a critical technique to accelerate the solution of linear systems, where performance heavily depends on the selection of preconditioning parameters. Traditional parameter selection approaches often define fixed constants for specific scenarios. However, they rely on domain expertise and fail to consider the instance-wise features for individual problems, limiting their performance. In contrast, machine learning (ML) approaches, though promising, are hindered by high inference costs and limited interpretability. To combine the strengths of both approaches, we propose a symbolic discovery framework-namely, **Sym**bolic **Ma**trix **P**reconditioning (**SymMaP**)-to learn efficient symbolic expressions for preconditioning parameters. Specifically, we employ a neural network to search the high-dimensional discrete space for expressions that can accurately predict the optimal parameters. The learned expression allows for high inference efficiency and excellent interpretability (expressed in concise symbolic formulas), making it simple and reliable for deployment. Experimental results show that SymMaP consistently outperforms traditional strategies across various benchmarks [1].

## 1 Introduction

Linear systems are foundational in machine learning, physics, engineering, and other scientific fields [31, 32, 15, 16]. Since analytical solutions are often unavailable, efficient numerical algorithms become essential [11]. Matrix preconditioning, a key technique in this domain, accelerates iterative solvers and improves computational stability [59, 7]. For instance, the successive over-relaxation (SOR) method optimizes convergence by integrating Gauss-Seidel iterations with a weighted update scheme governed by the over-relaxation factor $\omega$ [18].

The effectiveness of matrix preconditioning depends on key parameters, such as $\omega$ in SOR. Selecting $\omega > 1$ can accelerate convergence, while $\omega < 1$ may stabilize the process. This trade-off makes $\omega$ a critical parameter, directly influencing the performance of preconditioning.

Traditional parameter selection strategies often rely on domain expertise to define fixed constants for specific scenarios. However, (**challenge 1**) different problem parameters often require distinct optimal preconditioning parameters. Traditional strategies ignore instance-wise features—specific characteristics of individual problems, such as equation coefficients. This limits their adaptability to varying problem instances and tasks.

---

\*Corresponding author.

[1]Our code is available at https://github.com/Minghaom2/SymMaP.

39th Conference on Neural Information Processing Systems (NeurIPS 2025).

In contrast, machine learning (ML) approaches hold great promise but come with other challenges. First, (**challenge 2**) ML inference, while efficient on GPUs, performs poorly in CPU-only environments due to limited parallel processing capabilities. This is particularly problematic in linear system solver deployments, where GPU resources are often unavailable. Second, (**challenge 3**) the "black-box" nature of many ML techniques hinders a deeper understanding of the learned policies, raising concerns about their reliability.

In light of this, a natural solution is to combine the reliability and superior performance of these two paradigms. We propose a symbolic discovery framework—namely, **Sym**bolic **Ma**trix **P**reconditioning (**SymMaP**)—to learn efficient symbolic expressions for preconditioning parameters. The framework consists of three main steps. SymMaP first begins by applying a grid search to identify the optimal preconditioning parameters based on task-specific performance metrics. Next, the framework performs a risk-seeking search in the high-dimensional discrete space of symbolic expressions, evaluating the best-found symbolic expression using a risk-seeking strategy. Finally, these symbolic expressions can be directly integrated into the modern solvers for linear systems, significantly improving computational efficiency.

The key contributions and advantages of SymMaP are summarized as follows:

- We propose a symbolic discovery framework, SymMaP, to learn efficient symbolic expressions for preconditioning parameters.
- SymMaP exhibits excellent generalization, making it adaptable to a wide range of preconditioning methods and optimization objectives.
- The symbolic expressions derived by SymMaP are both interpretable and easy to integrate into solver environments, offering a practical and transparent approach to enhancing the performance of linear system solvers.

## 2 Preliminaries

### 2.1 Matrix Preconditioning Technique

Matrix preconditioning is a technique employed to accelerate the convergence of iterative solvers and enhance the stability of algorithms. It is generally employed in solving linear systems [7, 18], which are typically expressed in the form: $Ax = b$. The fundamental idea of preconditioning is to transform the original problem into an equivalent one with better numerical properties. Specifically, this technique involves finding a preconditioner $M$ that approximates either the inverse of $A$ or some form conducive to iterative solutions [7]. Consequently, the original equation is transformed into $MAx = Mb$. There are generally two optimization objectives: 1. To accelerate the convergence of iterations by altering the spectral distribution of the matrix $A$. 2. To reduce the condition number of the matrix $A$, thereby lessening its ill-conditioning and enhancing the stability of the iteration. Some common preconditioning techniques include the Jacobi, Gauss-Seidel, SOR [68], algebraic multigrid (AMG) [61], etc.

### 2.2 Prefix Notation

Prefix notation is a mathematical format where every operator precedes its operands, eliminating the need for parentheses required in conventional infix notation and simplifying symbolic manipulation. This representation is particularly advantageous in symbolic regression, as it allows mathematical expressions to be expressed as sequences of tokens that can be easily processed by neural networks.

In this notation, operators can be unary (e.g., sin, cos) or binary (e.g., $+$, $-$, $\times$, $\div$), while operands can be constants or variables [30]. Each prefix expression uniquely corresponds to a symbolic tree structure, facilitating the conversion back to the original mathematical expression [29].

The sequential nature of prefix notation aligns well with the architecture of recurrent neural networks (RNNs), which process information step by step. Unlike infix notation which may require variable-length look-ahead to determine the next valid token, prefix notation allows RNN to generate expressions through an auto-regressive process where each decision is well-defined based on previous tokens, and by removing the need for parentheses, it reduces the vocabulary size of possible tokens, which greatly enhances model training efficiency.

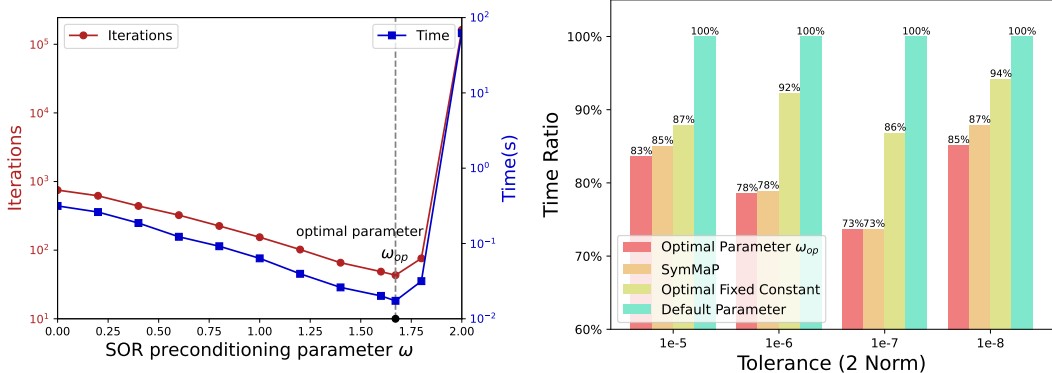

Figure 1: **Left.** Variation in iteration counts and computation times under different SOR preconditioning parameters applied to a linear system from a second-order elliptic PDE. **Right.** Ratio of average computation times at various tolerances to default parameter times under different SOR parameter selection schemes, evaluated on the second-order elliptic PDE dataset.

## 3 Motivation

The selection of matrix preconditioning parameters significantly affects their effectiveness [7]. To design appropriate algorithmic prediction parameters, we first analyze the optimization space for preconditioning parameter selection and investigate the existence of optimal parameters. Next, we analyze the unique challenges present in this scenario. Finally, to address these challenges, we design a symbolic discovery framework to select the parameters.

### 3.1 Motivation for Optimizing Preconditioning Parameters

As illustrated in Figure 1, the choice of relaxation factors $\omega$ significantly impacts the iteration count and computation time, when solving a second-order elliptic partial differential equation (PDE) [14] using SOR preconditioning [18]. There exists an optimal parameter $\omega_{op}$ that minimizes the computation time, with specific details available in the Appendix B.2.

To further analyze the optimization space of preconditioning parameters, we evaluate the impact of various parameter selection strategies on preconditioning performance. As shown in Figure 1, the 'Optimal Parameter $\omega_{op}$' represents the parameter that minimizes computation time in each experiment, serving as the theoretical upper limit of our optimization. The 'Optimal Fixed Constant' refers to a fixed constant that minimizes average computation time, and 'Default Parameter' corresponds to the default setting of $\omega = 1$ in the portable extensible toolkit for scientific computation (PETSc) [4]. The gap between the optimal fixed constant and the optimal parameter highlights significant potential for optimizing preconditioning parameter selection, motivating this paper. The performance of our SymMaP algorithm approaches the optimal parameter, demonstrating its accuracy in learning the optimal parameter expression.

### 3.2 Challenges in Predicting Efficient Preconditioning Parameters

We aim to develop a universal framework for predicting efficient parameters. However, the context of solving linear systems imposes specific challenges:

**(C1) Strong Generalization Capability**: Real-world scientific computing scenarios vary significantly. For instance, the choice of PDE grid form can lead to significant variations in matrix structure [25], resulting in distinct optimal parameters. Moreover, preconditioning addresses various optimization goals, such as reducing computational time, the number of iterations, and lowering condition numbers [7]. This necessitates that parameter prediction algorithms possess robust generalization capabilities: they should take problem scenarios and features as inputs while applying them to different preconditioning methods and optimization goals.

**(C2) Computational Efficiency**: linear system solver typically relies on Krylov subspace methods implemented in low-level libraries optimized for CPU architectures, such as PETSc [4], LA-

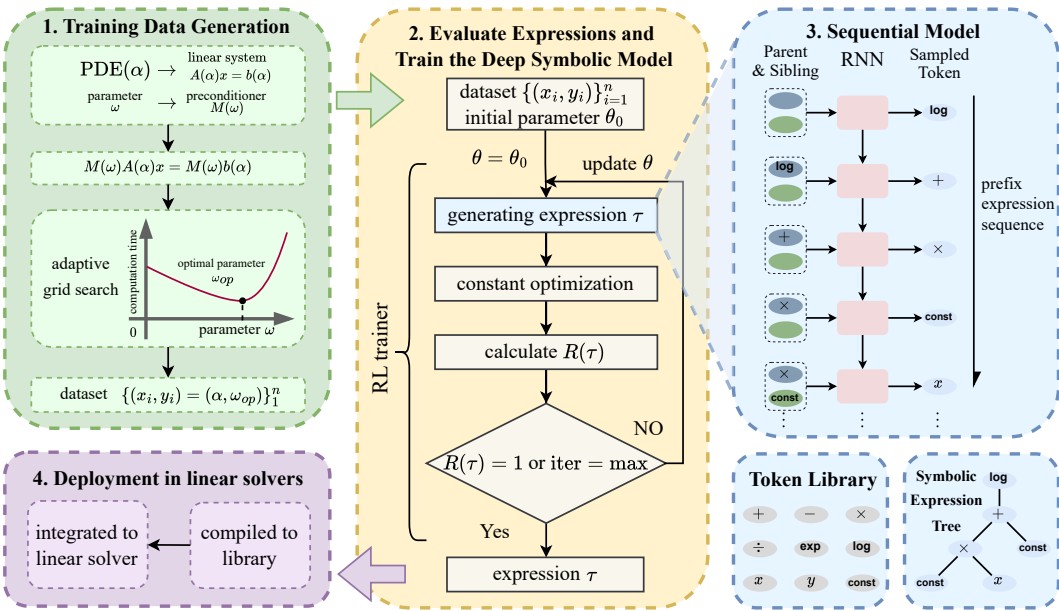

Figure 2: Illustration of how SymMaP discovers efficient symbolic expressions for preconditioning parameters. Part 1 demonstrates the acquisition of optimal parameters and dataset generation; Part 2 illustrates the training process of the RL-based deep symbolic discovery framework; Part 3 shows how the sequential model generates symbolic policies; Part 4 presents the deployment of symbolic expressions.

PACK [3]. Algorithms like generalized minimal residual method (GMRES) [52] and conjugate gradient (CG) [20] iteratively compute the matrix's invariant subspace, favoring single-threaded or limited multi-threaded execution modes. Preconditioning techniques aim to accelerate these solvers without significant additional computational overhead, often adopting implicit iterative formats (e.g., SOR [7]) or utilizing low-cost matrix decompositions (e.g., AMG [61]). Therefore, any parameter prediction algorithms must be compatible with CPU environments and seamlessly integrated into existing algorithm libraries. At the same time, it must maintain low computational costs to preserve the performance benefits of preconditioning.

**(C3) Algorithmic Transparency**: Algorithms in scientific computing often require rigorous analysis under mathematical theories. Opaque prediction algorithms could confuse researchers. For instance, the relaxation factor $\omega$ in SOR needs to avoid being too close to 0 or 2 in some scenarios [1]. This is an issue that opaque algorithms cannot avoid in advance. Moreover, interpretable algorithms can guide researchers to conduct further studies and reveal the underlying mathematical structures of problems. Therefore, these pose challenges to the transparency and interpretability of the parameter prediction algorithms.

### 3.3 Symbolic Discovery to Preconditioning Parameter Selection

Symbolic discovery extracts mathematical expressions from data, establishing relationships between problem features and optimal preconditioning parameters. Its integration into matrix preconditioning overcomes parameter selection challenges through a generalizable, efficient, and transparent approach.

Firstly, symbolic discovery can accommodate various types of input parameters and can specifically tailor symbolic expression learning for different preconditioning methods and optimization goals [10], thereby meeting the requirement for broad applicability in scientific computing tasks (**C1**). Secondly, the explicit expressions derived are computationally lightweight and can be quickly evaluated at runtime. They integrate seamlessly into existing CPU-based algorithm libraries like PETSc [4] with almost no overhead (**C2**). Thirdly, the symbolic discovery provides transparent and interpretable expressions [50], allowing researchers to understand the influence of parameters within existing theoretical frameworks and identify potential numerical stability issues. This interpretability fosters trust in the algorithm's predictions and supports further theoretical exploration (**C3**).

# 4 Method

This study focuses on enhancing the performance of parameterized preconditioners in solving linear systems derived from parameterized PDEs. Specifically, we investigate preconditioners with continuous parameters, such as the relaxation factor in SOR, while excluding those with discrete parameters like the level of fill-in ICC or ILU factorization.

We introduce a novel framework, SymMaP, for symbolic discovery in matrix preconditioning. As shown in Figure 2, we first obtain the optimal preconditioning parameters through a grid search to construct a training dataset. Then we employ an RNN to generate symbolic expressions in prefix notation, which are then evaluated for their fitness. The RNN is trained using a reward function based on the performance of the generated expressions. By optimizing the RNN parameters to maximize this reward function, we generate symbolic expressions that approximate the relationship between the problem's feature parameters (PDE parameters) and the optimal preconditioning parameters. Finally, we deploy the learned symbolic expressions into linear system solvers. The detailed steps are as follows and pseudocode is provided in the Appendix C.

## 4.1 Input Features and Training Data Generation

**Input Features**. In the context of solving parameterized PDEs, which frequently arise in linear systems, we consider feature parameters that characterize the equations. For instance, a second-order elliptic PDE can be expressed as: $a_{11}u_{xx} + a_{12}u_{xy} + a_{22}u_{yy} + a_1 u_x + a_2 u_y + a_0 u = f$, where the coefficients $a_{11}, a_{12}, a_{22}, \ldots$ represent the feature parameters of PDE (see Appendix D.1 for details). These feature parameters, denoted as $x_i$, serve as input features for the symbolic discovery process in SymMaP.

**Training Data Generation**. We assume that the preconditioning performance and parameters are continuous. For each linear system, we determine optimal preconditioning parameters through an adaptive grid search. Using SOR preconditioning as an example, we optimize the relaxation factor $\omega$ within [0, 2] to minimize computation time (or condition number). The search process involves: 1. Conducting an initial coarse grid search (step size: 0.01) to evaluate computation time (or condition number) for each $\omega$. 2. Identifying candidate regions with optimal performance. 3. Performing a refined grid search (step size: 0.001) within these regions.

This process yields the required training dataset, where each data point contains: 1. the problem feature parameters $x_i$. 2. the optimal preconditioning parameters $y_i$. $i = 1, 2, \ldots, n$, and $n$ is the number of data, typically set to $n = 1200$, with 1000 allocated for the training set and 200 for the test set.

## 4.2 The Generation of Symbolic Expressions

**Token Library**. For SymMaP, we define the library $\mathcal{L}$ of mathematical operators and operands as $\{+, -, \times, \div, \text{sqrt}, \exp, \log, \text{pow}, 1.0\}$. Although other operators such as poly, $\sin$ and $\cos$ are frequently used [62], we decided to exclude them because they offer limited explanatory power in matrix preconditioning and significantly increase the time and memory consumption during training.

After converting the mathematical expressions into prefix notation, we use this tokenized representation as a pre-order traversal of the expression tree [69]. In each iteration, the RNN receives a pair consisting of a parent node and a sibling node as inputs. Then the RNN outputs a categorical distribution over all possible next tokens. The parent node refers to the last incomplete operator that requires additional operands to form a complete expression. The sibling node, in the context of a binary operator, represents the operand that has already been processed and incorporated into the expression. In cases where no parent or sibling node is applicable, they are designated as empty nodes. This structured input method enables the RNN to maintain contextual awareness and effectively predict the sequence of tokens that form valid mathematical expressions.

**The Sequential Model**. During the generation of a single symbolic expression, the RNN emits a categorical distribution for each "next token" at each step. This distribution is represented as a vector $\psi_{\boldsymbol{\theta}}^{(i)}$, where $i$ denotes the $i$-th step and $\boldsymbol{\theta}$ represents the parameters of the RNN. The elements of the vector correspond to the probabilities of each token, conditioned on the previously selected tokens in

the traversal [46]:

$$\psi_{\boldsymbol{\theta}}^{(i)}(\boldsymbol{\tau}_i) = p(\boldsymbol{\tau}_i|\boldsymbol{\tau}_{1:i-1};\boldsymbol{\theta}). \tag{1}$$

Here, $\boldsymbol{\tau}_i$ denotes the index of the token selected at the $i$-th step. The probability of generating the entire symbolic expression $\boldsymbol{\tau}$ is then the product of the conditional probabilities of all tokens [46, 30]:

$$p(\boldsymbol{\tau}|\boldsymbol{\theta}) = \prod_{i=1}^{N} \psi_{\boldsymbol{\theta}}^{(i)}(\boldsymbol{\tau}_i). \tag{2}$$

**Optimization of Constants:** The library $\mathcal{L}$ incorporates a 'constant token,' which allows for the inclusion of various constant placeholders within sampled expressions. These placeholders serve as the parameters $\xi$ in the symbolic expression. We seek to find the optimal values of these parameters by maximizing the reward function: $\xi^* = \arg\max_\xi R(\tau;\xi)$, utilizing a nonlinear optimization method. This optimization is executed within each sampled expression as an integral part of computing the reward, prior to each training iteration.

## 4.3 The Reward Function

Once a symbolic expression is fully generated (i.e., the symbolic tree reaches all its leaf nodes), we evaluate its fitness by calculating the normalized root-mean-square error (NRMSE), a metric frequently used in genetic programming symbolic discovery [54]. It is defined as NRMSE $= \frac{1}{\sigma_y}\sqrt{\frac{1}{n}\sum_{i=1}^{n}(y_i - \hat{y}_i)^2}$, where $\hat{y}_i = \boldsymbol{\tau}(x_i)$ is the predicted value for the $i$-th sample, $x_i$ is the problem feature parameter, $y_i$ is the optimal preconditioning parameter, $\sigma_y$ is the standard deviation of the target values $y$, and $n$ is the number of data. To bound this fitness value between 0 and 1, we apply a squashing function: $R(\boldsymbol{\tau}) = 1/(1 + \text{NRMSE})$. Our objective is to maximize $R(\boldsymbol{\tau})$, thereby minimizing the NRMSE and improving the accuracy of the generated expressions.

## 4.4 The Training Algorithm

Although the objective function is well-defined, it is important to note that $R(\boldsymbol{\tau})$ is not a deterministic value but a random variable dependent on the RNN's parameters $\boldsymbol{\theta}$. Therefore, the key challenge is to establish an appropriate criterion for evaluating this random variable and then apply gradient-based optimization methods accordingly.

**Risk-seeking Policy.** It is natural to consider the expectation of the reward function, i.e., $\mathbb{E}_{\boldsymbol{\tau}\sim p(\boldsymbol{\tau};\boldsymbol{\theta})}[R(\boldsymbol{\tau})]$, as the objective function to optimize. We can apply the 'log-integral' trick [66] and obtain

$$\nabla_{\boldsymbol{\theta}}\, \mathbb{E}_{\boldsymbol{\tau}\sim p(\boldsymbol{\tau};\boldsymbol{\theta})}[R(\boldsymbol{\tau})] = \mathbb{E}_{\boldsymbol{\tau}\sim p(\boldsymbol{\tau};\boldsymbol{\theta})}[R(\boldsymbol{\tau})\nabla_{\boldsymbol{\theta}}\log p(\boldsymbol{\tau};\boldsymbol{\theta})]. \tag{3}$$

Thus, even though the expectation of the reward function is not directly differentiable with respect to $\boldsymbol{\theta}$, we can approximate the gradient using the sample mean.

In the context of symbolic regression, model performance is often driven by a few exceptional results that outperform others by a significant margin [46, 58]. With this in mind, we adopt a risk-seeking policy, which aims to maximize:

$$J(\boldsymbol{\theta},\varepsilon) = \mathbb{E}_{\boldsymbol{\tau}\sim p(\boldsymbol{\tau};\boldsymbol{\theta})}[R(\boldsymbol{\tau})|R(\boldsymbol{\tau}) > Q(\boldsymbol{\theta},\varepsilon)]. \tag{4}$$

Here, $\varepsilon$ is the risk factor, typically $\varepsilon = 0.05$, $Q(\boldsymbol{\theta},\varepsilon)$ is the $(1-\varepsilon)$-quantile of the reward distribution under parameter $\boldsymbol{\theta}$, i.e.

$$Q(\boldsymbol{\theta},\varepsilon) = \inf\{q \in \mathbb{R}|\text{CDF}(R(\boldsymbol{\tau});\boldsymbol{\theta}) \ge 1 - \varepsilon\}, \tag{5}$$

where $\text{CDF}(R(\boldsymbol{\tau});\boldsymbol{\theta})$ refers to the cumulative distribution function. From this, the gradient of $J(\boldsymbol{\theta},\varepsilon)$ can be derived as [46]:

$$\nabla_{\boldsymbol{\theta}}J(\boldsymbol{\theta},\varepsilon) = \mathbb{E}_{\boldsymbol{\tau}\sim p(\boldsymbol{\tau};\boldsymbol{\theta})}\left[\left(R(\boldsymbol{\tau}) - Q(\boldsymbol{\theta},\varepsilon)\right) \cdot \nabla_{\boldsymbol{\theta}}\log p(\boldsymbol{\tau};\boldsymbol{\theta})\Big| R(\boldsymbol{\tau}) > Q(\boldsymbol{\theta},\varepsilon)\right]. \tag{6}$$

This gradient can be estimated using Monte Carlo sampling:

$$\nabla_{\boldsymbol{\theta}}J(\boldsymbol{\theta},\varepsilon) \approx \hat{g} \triangleq \frac{1}{\varepsilon N}\sum_{i=1}^{N}(R(\boldsymbol{\tau}^{(i)}) - \tilde{Q}(\boldsymbol{\theta},\varepsilon))\nabla_{\boldsymbol{\theta}}\log p \cdot \mathbb{1}_{R(\boldsymbol{\tau}^{(i)})>\tilde{Q}(\boldsymbol{\theta},\varepsilon)}, \tag{7}$$

$\tilde{Q}(\boldsymbol{\theta},\varepsilon)$ is the empirical $(1-\varepsilon)$-quantile of the reward function. By concentrating on the top $\varepsilon$ percentile of samples, SymMaP emphasizes optimizing the best-performing solutions in preconditioning, thereby obtaining the optimal symbolic expressions for preconditioning parameters.

## 4.5 Deployment in Linear System Solver

After the training process, we obtained a symbolic expression for predicting the preconditioning parameter. The learned formula is exceptionally concise and incurs minimal computational cost. Therefore, we directly compile the learned policy into a lightweight shared object using a simple script and then integrate it into the linear system solver package (e.g., PETSc).

# 5 Experiments

We conducted comprehensive experiments to evaluate the SymMaP framework, organized into three primary sections: 1. Assessment of three different preconditioners and optimization goals across various datasets to determine the effectiveness of SymMaP, 2. Analysis of associated computational cost and the interpretability of the learned symbolic expressions, 3. Ablation studies.

**Preconditioners**: We considered three different preconditioners and various optimization metrics: 1. SOR preconditioner with the relaxation factor $\omega$ [18]; 2. SSOR preconditioner with the relaxation factor $\omega$ [18]; 3. AMG preconditioner with the threshold parameters $\theta_T$ [61].

**Datasets**: We investigated linear systems derived from five distinct PDE classes: 1. Darcy Flow Problems [35], 2. Second-order Elliptic PDEs [14], 3. Biharmonic Equations [5], 4. Thermal Problems [65]. 5. Poisson Equations [65]. All cases except biharmonic equations yield symmetric matrices. Notably, the non-symmetric matrices from biharmonic equations are incompatible with SSOR and AMG preconditioning techniques. Additionally, we conducted partial tests on the Lippmann-Schwinger equation (a numerical integration problem) and Markov chains (an optimization problem). Details of these experiments can be found in Appendix E.7.

**Baselines**: We compared SymMaP against various parameter selection methods for preconditioning. Specifically, the comparison involved the following scenarios: 1. No matrix preconditioning, 2. Default parameters in PETSc [4], 3. Fixed constants, 4. Optimized fixed constants.

**Experiment Settings**: All preconditioning procedures were uniformly implemented using the C-based PETSc library [4] to maintain evaluation consistency. The experiments were conducted within PETSc's linear solver framework (GMRES and CG) [18], with condition numbers computed through the built-in function `KSPComputeExtremeSingularValues`.

Details on preconditioners, the mathematical forms of datasets, tolerance metric (relative residual), and the runtime environment are available in Appendices B, D.1, D.3 and D.2, respectively. Information on the generation of training datasets for the following experiments and parameters of the SymMaP algorithm are outlined in Appendices D.4 and D.5. The generated dataset and training time are available in Appendix D.6. For an introduction to related work, see Appendix A.

## 5.1 Main Experiments

In these experiments, as shown in Tables 1, 2, 3, we optimized relaxation factors $\omega$ in both SOR and SSOR preconditioning, and threshold parameters $\theta_T$ in AMG preconditioning. For SOR and SSOR, we identified $\omega$ values that minimize computation time, forming the training dataset for SymMaP to learn symbolic expressions that optimize computational times for solutions. Similarly, for AMG, we

Table 1: Comparison of average computation times (seconds) for SOR with different $\omega$ selections, and tolerance is 1e-7. SymMaP 1 and 2 are the two learned expressions that achieved the highest reward function scores, with the best-performing method highlighted in bold.

| Dataset | Matrix size | No precondition | PETSc default $\omega = 1$ | Fixed constant $\omega = 0.2$ | Fixed constant $\omega = 1.8$ | Optimal constant | SymMaP 1 | SymMaP 2 |
|---|---|---|---|---|---|---|---|---|
| Biharmonic | $4.2 \times 10^3$ | 7.67 | 2.04 | 4.86 | 1.60 | 1.31 | **1.24** | 1.26 |
| Darcy Flow | $1.0 \times 10^4$ | 33.1 | 13.5 | 17.5 | 9.91 | 9.54 | **8.50** | 8.60 |
| Elliptic PDE | $4.0 \times 10^4$ | 31.3 | 21.0 | 21.4 | 17.5 | 16.6 | **15.8** | 16.3 |
| Poisson | $2.3 \times 10^3$ | $4.12 \times 10^{-2}$ | $1.95 \times 10^{-2}$ | $2.15 \times 10^{-2}$ | $1.95 \times 10^{-2}$ | $1.38 \times 10^{-2}$ | **$1.35 \times 10^{-2}$** | $1.36 \times 10^{-2}$ |
| Thermal | $2.8 \times 10^3$ | $2.23 \times 10^{-1}$ | $5.98 \times 10^{-2}$ | $2.07 \times 10^{-1}$ | $1.18 \times 10^{-1}$ | $5.94 \times 10^{-2}$ | **$5.76 \times 10^{-2}$** | $5.91 \times 10^{-2}$ |

Table 2: Comparison of average computation times (seconds) for SSOR with different $\omega$ selections, and tolerance is 1e-7. SymMaP 1 and 2 are the two learned expressions that achieved the highest reward function scores, with the best-performing method highlighted in bold.

| Dataset | Matrix size | No precondition | PETSc default $\omega = 1$ | Fixed constant $\omega = 0.2$ | Fixed constant $\omega = 1.8$ | Optimal constant | SymMaP 1 | SymMaP 2 |
|---|---|---|---|---|---|---|---|---|
| Darcy Flow | $4.9 \times 10^3$ | 4.18 | 0.488 | 0.757 | 1.09 | 0.448 | **0.412** | 0.523 |
| Elliptic PDE | $4.0 \times 10^4$ | 23.9 | 10.5 | 14.7 | 8.72 | 8.68 | **7.70** | 7.74 |
| Poisson | $2.3 \times 10^3$ | $2.12 \times 10^{-2}$ | $1.02 \times 10^{-2}$ | $1.93 \times 10^{-2}$ | $9.91 \times 10^{-3}$ | $9.89 \times 10^{-3}$ | **$9.10 \times 10^{-3}$** | $9.92 \times 10^{-3}$ |
| Thermal | $2.8 \times 10^3$ | $2.34 \times 10^{-1}$ | $2.69 \times 10^{-2}$ | $5.08 \times 10^{-2}$ | $9.87 \times 10^{-2}$ | $2.24 \times 10^{-2}$ | **$2.13 \times 10^{-2}$** | $2.14 \times 10^{-2}$ |

Table 3: Comparison of average condition numbers for preconditioned matrices using different threshold parameter $\theta_T$ selections in AMG. SymMaP 1 and 2 are the two learned expressions that achieved the highest reward function scores, with the best-performing method highlighted in bold.

| Dataset | Matrix size | No precondition | PETSc default $\theta_T = 0$ | Fixed constant $\theta_T = 0.2$ | Fixed constant $\theta_T = 0.8$ | Optimal constant | SymMaP 1 | SymMaP 2 |
|---|---|---|---|---|---|---|---|---|
| Darcy Flow | $1.0 \times 10^4$ | 752862 | 8204 | 19146 | 11426 | 7184 | **4824** | 5786 |
| Elliptic PDE | $4.0 \times 10^4$ | 6792 | 184.6 | 205.4 | 212.5 | 182.8 | **168.8** | 170.3 |
| Poisson | $1.0 \times 10^4$ | 1242 | 4.55 | 68.85 | 68.85 | 4.55 | **3.72** | 3.72 |
| Thermal | $2.8 \times 10^3$ | 7325 | 11.9 | 627.2 | 627.2 | 9.91 | **9.71** | 9.71 |

selected $\theta_T$ values that minimize the condition number of preconditioned matrices. Partial symbolic expressions can be found in Appendix E.1.

Experimental results indicate that SymMaP consistently outperforms all others across all experimental tasks. For SOR, Table 1 shows that SymMaP reduces computation times by up to 40% compared to PETSc's default settings and by 10% against the optimal constants. In SSOR, Table 2 shows that it cuts computation time and iteration counts by up to 27%, over PETSc's defaults, and achieves reductions of 11% in time compared to optimal constants. For AMG, Table 3 shows that SymMaP lowers the condition number by up to 40% relative to PETSc's defaults and 32% against the optimal constants. These results highlight SymMaP's ability to effectively derive high-performance symbolic expressions for various preconditioning parameters, showcasing its broad applicability and strong generalization across different preconditioning tasks.

Additionally, we analyze SymMaP's multi-parameter prediction capabilities, custom metric capabilities, generalization performance (e.g., performance outside the training parameter range and under varying matrix sizes, geometries), error reduction trajectories, and statistical characteristics of some experiments (e.g., medians, quartiles, etc.), which can be found in Appendices E.2, E.3, E.4, E.5, and E.6. One of the symbolic expressions derived by SymMaP exhibits significant advantages over existing learning-based algorithms in terms of applicability (as current learning-based methods often optimize only a single preconditioner), interpretability, and compatibility with CPU-based scenarios. Even without considering these advantages, we compared SymMaP with existing learning-based algorithms. SymMaP consistently demonstrated superior performance across all experiments, with detailed results provided in Appendix E.7.

## 5.2 Comparison with Neural Network Performance

To evaluate the deployment overhead and prediction performance of SymMaP, we compared it with a basic multi-layer perceptron (MLP) architecture. The MLP implementation consists of three fully connected layers, taking PDE parameters as input and generating preconditioning parameters as output. We employed ReLU activation functions and trained the model using mean squared error (MSE) between predicted and optimal parameters as the loss function. Both symbolic expression and MLP were executed in a CPU environment to simulate a modern solver environment.

As shown in Table 4, the runtime of symbolic expressions learned by SymMaP was only 20% of that of the MLP,

Table 4: Comparison of the runtime required for symbolic expression and MLP to predict the SOR relaxation factor and the subsequent average solution time, using the Darcy flow dataset with a matrix size of $10^3$ and tolerance is 1e-5.

| | Runtime (s) | Solution time (s) |
|---|---|---|
| MLP | $5.1 \times 10^{-5}$ | $7.1 \times 10^{-1}$ |
| Symbol | $1.1 \times 10^{-5}$ | $7.1 \times 10^{-1}$ |

primarily due to the poor performance of neural networks in a pure CPU environment, highlighting SymMaP's computational efficiency. Furthermore, the average solution times for parameters predicted by both symbolic expressions and MLP were closely matched. This demonstrates that symbolic expressions possess equivalent expressive capabilities to neural networks in this scenario, effectively approximating the optimal parameter expressions.

## 5.3 Interpretable Analysis

In Table 5, we report a subset of the learned symbolic expressions, with the mathematical significance of the related symbols detailed in Appendix E.8. More symbolic expressions can be found in Appendix E.1. These symbolic expressions are notably more concise and selective, not utilizing all candidate parameters and symbols, which aids researchers in analyzing their underlying relationships.

Table 5: Partial symbolic expressions learned from some of the main experiments.

| | Dataset | Symbolic expression |
|---|---|---|
| SOR | Biharmonic | $1.0 + 1.0/(4.0 + 1.0/x_2) + 1.0/x_1$ |
| SOR | Elliptic PDE | $1.0 + 1.0/(x_2 + 1.0 + 1.0/(x_2 + 4.0))$ |
| SOR | Darcy Flow | $1.0 + 1.0/(x_4 + 1.0)$ |
| SSOR | Elliptic PDE | $1.0 + 1.0/(x_2 + 1.2)$ |
| AMG | Elliptic PDE | $(x_1 x_3 + 1)/7$ |

For instance, in the context of SOR and SSOR preconditioning, empirical evidence suggests that smaller relaxation factors should be chosen when diagonal components are relatively small. Our experimental findings corroborate this: for the second-order elliptical PDE dataset, the symbolic expressions derived for SOR and SSOR preconditioning depend solely on $x_2$, with larger $x_2$ values leading to smaller predicted relaxation factors, exemplified by $1.0 + \frac{1.0}{(x_2+1.2)}$. Here, $x_2$ represents the coupling coefficient of the elliptical PDE, which directly influences the relative size of the non-diagonal components of the generated matrix, whereas other coefficients have minimal impact. As the coupling coefficient increases, the relative numerical value of the non-diagonal components increases, and the diagonal components reduce correspondingly, aligning with empirical observations.

These experimental outcomes demonstrate that SymMaP can derive interpretable and efficient symbolic expressions for parameters, further aiding researchers in understanding and exploring the underlying mathematical principles.

## 5.4 Ablation Experiments

We conducted an ablation study using SymMaP to evaluate the impact of different mathematical operator selections, as described in Table 6. In the main experiments, We utilized the operator set $\{+, -, \times, \div, \text{sqrt}, \exp, \log, \text{pow}, 1.0\}$ listed in the second row.

The results indicate that this selection of operators achieves a balance between predictive performance and training time efficiency, meeting our expectations. A detailed hyperparameter sensitivity analysis concerning learning rate, batch size, and dataset size is provided in Appendix E.9. Furthermore, comparative experiments against alternative parameter search methods and other interpretable algorithms are presented in Appendices E.10 and E.11.

Table 6: Comparison of the impact of the choice of mathematical operators on preconditioning and training time. The first column lists the selected operators, the second column shows the condition numbers of preconditioned matrices derived from AMG parameter predictions on the Darcy flow dataset (lower is better), and the third column displays SymMaP training times.

| Functionset | Condition number | time(s) |
|---|---|---|
| $+, -, \times, \div, \text{poly}$ | 6803.8 | 15351 |
| $+, -, \times, \div, \text{sqrt}, \exp, \log, \text{pow}, 1.0$ | 7086.9 | 703.17 |
| $+, -, \times, \div, \text{sqrt}, \exp, \log, \sin, \cos, \text{pow}, 1.0$ | 7172.6 | 635.82 |
| $+, -, \div, 1.0, \text{pow}$ | 7241.8 | 703.26 |
| $+, -, \times, \div, \text{sqrt}, \text{pow}, 1.0$ | 7271.1 | 746.80 |
| $+, -, \times, \div, \text{pow}, 1.0$ | 7301.4 | 702.46 |

# 6 Limitations and Conclusions

In this paper, we propose SymMaP, a deep symbolic discovery framework designed for predicting efficient matrix preconditioning parameters. Experiments show that SymMaP can predict high-performance parameters and is applicable across a variety of preconditioning and optimization objectives. Additionally, SymMaP is easy to deploy with virtually no computational cost.

Future work will focus on optimizing preconditioning for specific matrix structures, such as symmetric and upper triangular matrices. We also aim to analyze the mathematical significance of the learned symbolic expressions from a theoretical perspective, such as exploring the impact of problem features on the solution process through pseudospectral analysis. Furthermore, plans to extend SymMaP to additional preconditioning methods (e.g., ILU, ICC) are underway. We are confident in the symbolic model's immense potential for broad real-world applications, especially in matrix preconditioning.

## Acknowledgements

The authors would like to thank all the anonymous reviewers for their insightful comments and valuable suggestions. This work was supported by the National Key R&D Program of China under contract 2022ZD0119801, and the National Nature Science Foundations of China grants U23A20388, 62021001.

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

# A   Related work

## A.1   Machine Learning for Algorithm Discovery

Machine learning has the potential to uncover implicit rules beyond human intuition from training data, enabling the construction of algorithms that outperform handcrafted programs. Approaches to algorithm discovery in machine learning encompass symbolic discovery, program search, and more. Specifically, program search focuses on optimizing the computational processes of algorithms. For example, [44] explores the discovery of faster sorting algorithms, while [8] investigates efficient optimization algorithms.

In contrast, symbolic discovery aims to search within the space of small mathematical expressions rather than computational streams [46, 30]. This approach is analogous to an extreme form of model distillation, where knowledge extracted from black-box neural networks is distilled into explicit mathematical expressions. Traditional methods for symbolic discovery have relied on evolutionary algorithms, including genetic programming [47]. Recently, deep learning has emerged as a powerful tool in this domain, offering enhanced representational capacity and new avenues for solving symbolic discovery problems [54, 10].

## A.2   Neural Networks for Matrix Preconditioning

Recent studies have explored the use of neural networks to improve matrix preconditioning techniques. [21, 42, 57, 67, 65, 12, 41, 36, 37, 38, 48, 28, 39, 33, 22, 43] demonstrate the effectiveness of neural networks in refining multigrid preconditioning algorithms, thus streamlining the computational process. [19] utilized Convolutional Neural Networks (CNNs) for the optimization of block Jacobi preconditioning algorithms, while [56] developed corresponding Incomplete Lower-Upper Decomposition (ILU) preconditioning algorithms leveraging machine learning insights. Although these algorithms achieved impressive results, they still face challenges such as limited interpretability and reduced computational efficiency when deployed in pure CPU environments. This paper attempts to address these issues by incorporating symbolic discovery into the framework.

# B   Detailed introduction of matrix preconditioning

## B.1   Overview of Matrix Preconditioning Methods

- **Jacobi Method**: The Jacobi preconditioner utilizes only the diagonal elements of a matrix to precondition a linear system. By approximating the inverse of the diagonal matrix, this method is computationally simple and effective for systems with strong diagonal dominance. However, its convergence rate can be slow, and its performance diminishes for poorly conditioned or weakly diagonally dominant matrices. The Jacobi method is typically used as a baseline for comparison with more sophisticated preconditioners [53].

- **Gauss-Seidel (GS) Method**: The Gauss-Seidel preconditioner improves upon the Jacobi method by considering both the lower triangular and diagonal parts of the matrix in a sequential manner. Unlike the Jacobi method, which updates all variables simultaneously, the GS method updates each variable in sequence using the most recent values. This leads to faster convergence, especially for diagonally dominant matrices. However, the GS method can still struggle with poorly conditioned systems, and its forward-only approach can limit performance in some applications [53].

- **Successive Over-Relaxation (SOR)**: The SOR method builds on the Gauss-Seidel method by introducing a relaxation factor $\omega$ to accelerate convergence. This factor allows for over-relaxation ($\omega > 1$) or under-relaxation ($\omega < 1$), tuning the method for faster performance on certain types of problems. SOR can significantly reduce the number of iterations needed for convergence compared to both the Jacobi and GS methods, but choosing the optimal relaxation factor is problem-dependent [68].

- **Symmetric Successive Over-Relaxation (SSOR)**: SSOR is a symmetric version of the SOR method, where relaxation is applied in both forward and backward sweeps of the matrix. This bidirectional process improves stability and is well-suited for use with iterative solvers like the conjugate gradient method, which requires symmetric preconditioners. SSOR's

symmetry ensures that the preconditioner maintains the properties needed for efficient and stable convergence, making it a popular choice for symmetric positive-definite systems [18].

- **Algebraic Multigrid (AMG)**: AMG is an advanced preconditioning technique designed to handle large, sparse systems of linear equations, especially those arising from the discretization of partial differential equations. Unlike traditional methods, AMG operates on multiple levels of the matrix structure, coarsening the matrix to form a hierarchy of smaller systems that are easier to solve. Solutions on the coarser grids are then interpolated back to the finer grids. This multilevel approach makes AMG highly efficient for large-scale problems, as it can dramatically reduce the number of iterations needed to achieve convergence. AMG is often used in combination with methods like SSOR or Gauss-Seidel as a smoother on each grid level, and it is particularly effective in cases where the problem exhibits a multiscale nature [51].

**Relationship Among Jacobi, GS, and SOR Methods**: The Jacobi method is the simplest of the three, using only diagonal information. The GS method improves upon the Jacobi method by using both diagonal and lower triangular matrix elements to achieve faster convergence. SOR further refines the GS method by introducing a relaxation factor to optimize the update process. Both the GS and SOR methods can be seen as iterative improvements on the Jacobi method, with SOR offering a more flexible and potentially faster alternative by adjusting the relaxation factor. SSOR extends SOR symmetrically, making it suitable for use in more advanced iterative solvers like the conjugate gradient method [53, 18].

## B.2 Parameters in Matrix Preconditioning

The choice of preconditioning parameters significantly influences the effectiveness of the preconditioning process, especially in the iterative solving of linear systems [7]. Below, we discuss three specific preconditioning techniques—SOR, SSOR, and AMG—focusing particularly on how their key parameters affect the preconditioning results.

### B.2.1 Relaxation Factor $\omega$ in SOR and SSOR Methods

In the SOR preconditioning method, the relaxation factor $\omega$ is a critical parameter that determines the acceleration of iteration. SOR evolves from the Gauss-Seidel method by introducing $\omega$ to speed up convergence. The SOR iteration formula is given by:

$$\boldsymbol{x}^{(k+1)} = (\boldsymbol{D} + \omega\boldsymbol{L})^{-1}\left[(1-\omega)\boldsymbol{D}\boldsymbol{x}^{(k)} + \omega\boldsymbol{b} - \omega\boldsymbol{U}\boldsymbol{x}^{(k)}\right], \qquad (8)$$

where $\boldsymbol{D}$, $\boldsymbol{L}$, and $\boldsymbol{U}$ are the diagonal, strictly lower triangular, and strictly upper triangular parts of the matrix $\boldsymbol{A}$, respectively [18].

The SSOR preconditioning method can be represented by the following formula:

$$\boldsymbol{M}_{\text{SSOR}} = \frac{1}{\omega(2-\omega)}(\boldsymbol{D} - \omega\boldsymbol{U})\boldsymbol{D}^{-1}(\boldsymbol{D} - \omega\boldsymbol{L}), \qquad (9)$$

where $\boldsymbol{M}_{\text{SSOR}}$ constitutes the preconditioner, and $\boldsymbol{D}$, $\boldsymbol{L}$, $\boldsymbol{U}$, and $\omega$ are defined similarly to their roles in the SOR method. This symmetrical formulation enhances the stability and effectiveness of the preconditioning, particularly benefiting symmetric positive-definite matrices by optimizing the convergence properties of the iterative solver [18].

The choice of $\omega$ directly impacts the speed of convergence and the condition number of the matrix. Different problems and scenarios often require different choices of $\omega$, which typically need to be determined based on the specific properties of the problem and through numerical experimentation [18]. In the PETSc library, the default relaxation factor $\omega$ for both SOR and SSOR is set to 1, at which point SOR degenerates to GS preconditioning.

### B.2.2 Threshold Parameters $\theta_T$ in AMG

In the AMG method, the threshold parameter $\theta_T$ determines whether the non-zero elements of the matrix are "strong" enough to be considered in the construction of a coarse grid during the multigrid

process. This parameter is crucial for establishing the connectivity between coarse and fine grids in the hierarchical multilevel structure [51].

The AMG method solves the equation system through multiple levels of grids, each corresponding to a coarser version of the original problem. During this process, the threshold parameter is used to determine whether a given non-zero matrix element is strong enough to keep the corresponding grid points connected during coarsening.

- A lower threshold often leads to more elements being considered as strong connections, which might increase the complexity of the coarse grid but can help preserve the essential features of the original problem, thus improving the efficiency and convergence of the multigrid method.

- A higher threshold might result in fewer strong connections, thereby reducing the complexity of the coarse grid. However, this can weaken the effectiveness of the AMG method, especially in maintaining the features of the original problem.

Different values of $\theta_T$ directly influence the condition number of the preconditioned matrix. Selecting the appropriate threshold parameter typically involves considering the specific structure and features of the problem, and adjustments are made through experimental fine-tuning to achieve the optimal balance [61]. In the PETSc library, the default threshold parameter $\theta_T$ is set to 0.

# C Algorithm Pseudocode

---

**Algorithm 1:** RNN-based Symbolic Discovery Process

---

**Input:** RNN with parameter $\boldsymbol{\theta}$, the library of tokens $\mathcal{L}$

$\boldsymbol{\tau} \leftarrow [\,]$

parent$(0)$, sibling$(0) \leftarrow$ empty node

$x_0 \leftarrow$ parent$(0)$||sibling$(0)$     // $x$ is the concatenation of parent and sibling nodes

$h_0 \leftarrow 0$     // Initialize hidden state of RNN

**for** $t = 1, 2, \cdots$ **do**

    $(\psi_t, h_t) \leftarrow \text{RNN}(x_{t-1}, h_{t-1}; \boldsymbol{\theta})$   // $\psi_t$ is the categorical distribution of the next token

    $\psi_t \leftarrow \text{ApplyConstraint}(\psi_t, \mathcal{L}, \boldsymbol{\tau})$     // Regularize the distribution

    Sample token $\boldsymbol{\tau}_t \sim \psi_t$

    **if** $Arity(\boldsymbol{\tau}_t) > 0$ **then**

        // Arity$(\boldsymbol{\tau}_i)$ denotes the number of operands of $\boldsymbol{\tau}_i$

        parent$(t) \leftarrow \boldsymbol{\tau}_t$

        sibling$(t) \leftarrow$ empty node

    **else**

        // When Arity$(\boldsymbol{\tau}_t) = 0$, go back to the last incomplete operator node

        count $\leftarrow 0$

        **for** $i = t, t-1, \ldots, 1$ **do**

            // Backward iteration

            count $\leftarrow$ count + Arity$(\tau_i) - 1$

            **if** *count = 0* **then**

                parent$(t) \leftarrow \boldsymbol{\tau}_i$

                sibling$(t) \leftarrow \boldsymbol{\tau}_{i+1}$

                **break**

        **if** *count = −1* **then**

            // The expression sequence is complete

            **break**

    $x_t \leftarrow$ parent$(t)$||sibling$(t)$

**Output:** Prefix expression sequence $\boldsymbol{\tau}$

---

---

**Algorithm 2:** Deep Symbolic Optimization for Matrix Preconditioning Parameter

---

**Input** : RNN with initial parameter $\boldsymbol{\theta}_0$, the library of tokens $\mathcal{L}$, batch size $N$, iteration number $J$, risk factor $\varepsilon$, and learning rate $\alpha$

$\boldsymbol{\theta} \leftarrow \boldsymbol{\theta}_0$

$j \leftarrow 0$

**repeat**

    **for** $i = 1, 2, \ldots, N$ **do**

        $\boldsymbol{\tau}^{(i)} \leftarrow \text{SymbolicDiscover}(\boldsymbol{\theta}, \mathcal{L})$

        $\xi^* \leftarrow \arg\max\{\xi \text{ in } \boldsymbol{\tau} \text{ as constant placeholder} : R(\boldsymbol{\tau}; \xi)\}$     // Constant optimization

        $\boldsymbol{\tau}^{(i)} \leftarrow \text{ReplaceConstant}(\boldsymbol{\tau}^{(i)}, \xi^*)$

        Compute $\hat{g}_1$ using $\boldsymbol{\tau}^{(i)}$ and $\boldsymbol{\theta}$     // See Eq. (7)

        Compute $\hat{g}_2$ as entropy gradient

        $\boldsymbol{\theta} \leftarrow \boldsymbol{\theta} + \alpha(\hat{g}_1 + \hat{g}_2)$     // Update the parameter

        Train model: update $p_{\boldsymbol{\theta}}$ via PPO by optimizing $J(\boldsymbol{\theta}; \epsilon)$

**until** $j = J$ *or convergence*

**Output :** The best symbolic expression $\boldsymbol{\tau}^*$

---

# D Experiment Settings

## D.1 Datasets

### 1. Darcy Flow Problem

We consider two-dimensional Darcy flows, which can be described by the following equation [35, 49, 27, 40]:

$$-\nabla \cdot (K(x,y)\nabla h(x,y)) = f, \tag{10}$$

where $K$ is the permeability field, $h$ is the pressure, and $f$ is a source term which can be either a constant or a space-dependent function.

In our experiment, $K(x,y)$ is generated using truncated Chebyshev polynomials. We convert the Darcy flow problem into a system of linear equations using the central difference scheme of Finite Difference Methods (FDM) [32]. The coefficients of the Chebyshev polynomials serve as input features for our symbolic discovery framework.

### 2. Second-order Elliptic Partial Differential Equation

We consider general two-dimensional second-order elliptic partial differential equations, which are frequently described by the following generic form [14, 6]:

$$\mathcal{L}u \equiv a_{11}u_{xx} + a_{12}u_{xy} + a_{22}u_{yy} + a_1u_x + a_2u_y + a_0u = f, \tag{11}$$

where $a_0, a_1, a_2, a_{11}, a_{12}, a_{22}$ are constants, and $f$ represents the source term, depending on $x, y$. The variables $u, u_x, u_y$ are the dependent variables and their partial derivatives. The equation is classified as elliptic if $4a_{11}a_{22} > a_{12}^2$.

In our experiments, $a_{11}, a_{22}, a_1, a_2, a_0$ are uniformly sampled within the range $(-1, 1)$, while the coupling term $a_{12}$ is sampled within $(-0.01, 0.01)$. We then select equations that satisfy the elliptic condition to form our dataset. Similar to the approach with the Darcy flow problem, we convert the PDE into a system of linear equations using the central difference scheme of FDM. The coefficients $a_0, a_1, a_2, a_{11}, a_{12}, a_{22}$ serve as input features for our symbolic discovery framework. When discussing SSOR preconditioning, we set $a_1$ and $a_2$ to zero to ensure the resulting matrix remains symmetric.

### 3. Biharmonic Equation

We consider the biharmonic equation, a fourth-order elliptic equation, defined on a domain $\Omega \subset \mathbb{R}^2$. The equation is expressed as follows [9, 17, 5]:

$$\nabla^4 u = f \quad \text{in } \Omega = [0, a] \times [0, b], \tag{12}$$

where $\nabla^4 \equiv \nabla^2\nabla^2$ represents the biharmonic operator and $f = 4.0\pi^4 \sin(\pi x)\sin(\pi y)$ is the prescribed source term.

In our experiments, we construct the dataset by varying the solution domain $\Omega = [0, a] \times [0, b]$. We utilize the discontinuous Galerkin finite element method from the FEniCS library to transform this problem into a system of linear equations [5]. The parameters $a, b$ of the domain serve as input features for our symbolic discovery framework.

### 4. Poisson Equation

We consider a two-dimensional Poisson equation, which can be described by the following equation [65, 24, 70]:

$$\nabla^2 u = f \quad \text{in } \Omega = [0, 1]^2. \tag{13}$$

Physical Contexts in which the Poisson Equation Appears: 1. Electrostatics; 2. Gravitation; 3. Fluid Dynamics.

In our experiments, we address the Poisson equation within a square domain , where both the boundary conditions on all four sides and the source term $f$ on the equation's left-hand side are generated using third-order truncated Chebyshev polynomials. The finite difference method with a central difference scheme is employed to discretize the equation into a linear system. The Chebyshev coefficients serve as parameters for our symbolic discovery framework [13].

## 5. Thermal Problem

We consider a two-dimensional thermal steady state equation, which can be described by the following equation [65, 55, 26]:

$$\frac{\partial^2 T}{\partial x^2} + \frac{\partial^2 T}{\partial y^2} = 0, \tag{14}$$

where $T$ is the temperature. We examine the steady-state thermal equation in thermodynamics. As with the previous equation, we still solve this equation in the square domain. The boundary temperatures on the left and right boundaries are determined by random values ranging from -100 to 0 and 0 to 100, respectively. The top and bottom boundary temperature functions are generated by third-order truncated Chebyshev polynomials. The boundary temperature and the coefficients of the Chebyshev polynomials serve as parameters for our symbolic discovery framework.

## 6. Lippmann-Schwinger Equation

We consider the 2D Lippmann-Schwinger integral equation describing quantum scattering processes [63, 2]:

$$\psi(\boldsymbol{r}) = \phi(\boldsymbol{r}) + \int G_0(\boldsymbol{r}, \boldsymbol{r}')V(\boldsymbol{r}')\psi(\boldsymbol{r}')d\boldsymbol{r}', \tag{15}$$

where $\phi(\boldsymbol{r})$ is the incident wavefunction and $G_0$ denotes the free-particle Green's function. For symmetric potentials, we parameterize $V(r)$ using Chebyshev polynomials on a radial grid $r \in [0, R_{max}]$:

$$V(r) = \sum_{k=0}^{N} c_k T_k \left( \frac{2r}{R_{max}} - 1 \right). \tag{16}$$

Through Galerkin discretization with Chebyshev basis functions, we convert the integral equation into a sparse linear system:

$$[\boldsymbol{I} - \boldsymbol{G}_0 \boldsymbol{V}] \boldsymbol{\psi} = \boldsymbol{\phi}, \tag{17}$$

where $\boldsymbol{G}_0$ contains integrated Green's function matrix elements and $\boldsymbol{V}$ is the diagonal potential matrix constructed from Chebyshev coefficients. This formulation enables efficient computation of scattering cross-sections across different potential configurations. The Chebyshev coefficients $\{c_k\}$ serve as parameters for our symbolic discovery framework.

## 7. Markov chain (Boltzmann-Distributed Brownian Motion)

We consider a Markov chain optimization problem governed by the steady-state 1D Fokker-Planck equation with a Chebyshev-parameterized potential energy field [45, 64]:

$$\nabla \cdot (D\nabla\rho + \rho\nabla\Phi(x)) = f, \tag{18}$$

where $\Phi(x) = \sum_{k=0}^{N} c_k T_k(x)$ is the potential energy defined by Chebyshev polynomials $T_k$, $D$ is the diffusion coefficient, and $f$ is a source term. The equilibrium distribution follows $\rho_{eq} \propto e^{-\Phi(x)/k_B T}$.

In our experiments, the potential field $\Phi(x)$ is generated using truncated Chebyshev polynomials with coefficients $\{c_k\}$ sampled uniformly from $[-1, 1]$. We discretize the Fokker-Planck equation into a sparse, nonsymmetric linear system $A(\{c_k\})\boldsymbol{\rho} = \boldsymbol{f}$ via spectral collocation methods with Chebyshev basis functions. The Chebyshev coefficients $\{c_k\}$ serve as parameters for our symbolic discovery framework.

### D.2 Environment

To ensure consistency in our evaluations, all comparative experiments were conducted under uniform computing environments. Specifically, the environments used are detailed as follows:

1. Environment (Env1):
   - Platform: Windows11 version 22631.4169, WSL
   - Operating System: Ubuntu 22.04.3
   - CPU Processor: AMD Ryzen 9 5900HX with Radeon Graphics CPU, clocked at 3.30GHz
2. Environment (Env2):
   - Platform & Operating System: Ubuntu 18.04.4 LTS

- CPU Processor: Intel(R) Xeon(R) Gold 6246R CPU at 3.40GHz
- GPU Processor: GeForce RTX 3090 24GB
- Library: CUDA Version 11.3

Speed tests for solving linear systems were performed in Env 1, while all training related to symbolic discovery was conducted in Env 2.

### D.3 Tolerance Metric for Linear Systems

- Relative Residual:
  To estimate the relative accuracy of the solution $\tilde{x}$ for a linear system $Ax = b$, we employ the relative residual as follows:

$$\text{Relative Residual} = \frac{||A\tilde{x} - b||_2}{||b||_2}. \tag{19}$$

  Here, $\tilde{x}$ represents the solution predicted by the model, and $b$ is the right-hand side vector of the linear system. When $\tilde{x}$ is the exact solution, the Relative Residual equals 0.

### D.4 Training Data Generation

We employed an adaptive grid search to generate the training dataset. Initially, we traversed a coarse grid, sampling every 0.05, and from this dataset, we selected the three points with the smallest values. Subsequently, we conducted a finer grid search around these points, sampling every 0.001, to identify the point with the minimum value, which we designated as our optimal parameter. Particularly, after experimental validation confirmed the dataset's convexity, we utilized a binary search sampling method for a dataset derived from the second-order elliptic equation's SOR preconditioning. Starting with points at 0.0, 1.0, and 2.0, we compared these values. If the value at 0.0 was lowest, we computed at 0.5; if at 2.0, then at 1.5; and if at 1.0, then at both 0.5 and 1.5. This process was repeated until a minimum point was achieved with a precision of 0.001.

For SOR preconditioning, we evaluated second-order elliptic equations, Darcy flow equations, and biharmonic equations, with solution time as the metric for optimal preprocessing parameters, achieved by minimizing solution time using the previously described grid method. In SSOR preconditioning, applied to second-order elliptic and Darcy flow equations, we utilized a hybrid metric that combined normalized computation time and iteration counts, aiming to simultaneously optimize both iteration counts and solution times. For AMG preconditioning, also examined with second-order elliptic and Darcy flow equations, we used the condition number of the preconditioned matrix as the metric, where a lower value indicates better performance.

### D.5 Parameters of SymMAP

**Experimental Setup**. SymMAP is implemented using the LSTM architecture with one layer and 32 units. More details about the hyperparameters are provided in Table 7.

Table 7: Hyperparameters of SymMAP (Default Model)

| Hyperparameter | Value |
|---|---|
| Number of LSTM layers | 1 |
| Number of LSTM units | 32 |
| Number of training samples | 2,000,000 |
| Batch size | 1,000 |
| Risk factor $\varepsilon$ | 0.05 |
| Minimal expression length | 4 |
| Maximal expression length | 64 |
| Learning rate | 0.0005 |
| Weight of entropy regularization | 0.03 |

**Restricting searching space**. We employ specific constraints within our framework to streamline the exploration of expression spaces effectively and ensure they remain within practical and manageable bounds:

1. **Bounds on expression length.** To strike a balance between complexity and manageability, we set boundaries for expression lengths: a minimum of 4 and a maximum of 64 characters. This ensures that expressions are neither overly trivial nor excessively complicated.

2. **Constant combination.** We restrict expressions such that the operands of any binary operator are not both constants. This is out of the simple intuition that, if both operands are constants, the combination of the two can be precomputed and replaced with a single constant.

3. **Inverse operator exclusion.** We preclude unary operators from having their inverses as children to avoid redundant computations and meaningless expressions, such as in $\log(\exp(x))$.

4. **Trigonometric Constraints.** Expressions involving trigonometric operators should not include descendants within their formulation. For instance, $\sin(x + \cos(x))$ is restricted because it combines trigonometric operators in a way that is uncommon in scientific contexts.

## D.6 Computational Time for Related Algorithms

- **Dataset Generation Time:**
    - Darcy Flow Problem: 40 hours
    - Second-order Elliptic Partial Differential Equation: 40 hours
    - Biharmonic Equation: 100 hours
    - Poisson Equation: 6 hours
    - Thermal Problem: 6 hours

- **SymMAP Training Time:** For each run, 1000 iterations are performed.
    - Without polynomials in the Token Library: approximately 800 seconds.
    - With polynomials in the Token Library: approximately 2600 seconds.

- **MLP Training Time (in Experiment 5.2):**
    - Training until full convergence: 1.5 hours.

# E  Experimental Data and Supplementary Experiments

## E.1  Symbolic Expressions from Main Experiments

This section documents some of the learned expressions from the main experiments, corresponding to "SymMaP1" in Tables 1, 2, and 3.

- Second-order elliptic PDE problem, AMG preconditioning:

$$\frac{x_1 x_3 + 1.0}{x_1 + 7.0}$$

  Parameter meanings: $x_1$-$x_6$ represent the coefficients $a_{11}$, $a_{12}$, $a_{22}$, $a_1$, $a_2$, and $a_0$ in the second-order elliptic equation.

- Second-order elliptic PDE problem, SSOR preconditioning:

$$(2x_2 - \frac{1}{4x_1 + 2x_3})(-0.21785x_1^3 - 63.6118x_1^2 x_2 + 0.206541x_1^2 x_3 - 0.235667x_1^2 x_4$$
$$+ 0.269472x_1^2 - 967.517x_1 x_2^2 - 61.2291x_1 x_2 x_3 + 1.68205x_1 x_2 x_4 + 5.07925x_1 x_2$$
$$-0.0221322x_1 x_3^2 - 0.454257x_1 x_3 x_4 - 0.0693756x_1 x_3 + 0.411528x_1 x_4^2 + 0.0311608x_1 x_4$$
$$- 7.53439x_1 + 9506.4x_2^3 - 468.735x_2^2 x_3 - 154.885x_2^2 x_4 + 410.223x_2^2 - 25.5913x_2 x_3^2$$
$$+7.92627x_2 x_3 x_4 + 5.30828x_2 x_3 + 3.82512x_2 x_4^2 - 7.0487x_2 x_4 - 0.612507x_2 - 0.180432x_3^3$$
$$-0.0462734x_3^2 x_4 + 0.310649x_3^2 + 0.257121x_3 x_4^2 - 0.0962336x_3 x_4 - 3.86012x_3 + 0.13906x_4^3$$
$$- 0.389893x_4^2 + 0.3144x_4 - 0.0111835)$$

  Parameter meanings: $x_1$-$x_4$ represent the coefficients $a_{11}$, $a_{12}$, $a_{22}$, and $a_0$ in the second-order elliptic equation.

- Darcy flow problem, SSOR preconditioning:

$$\frac{1.0}{x_1(x_{14} + x_4) + 1.0}$$

  Parameter meanings: $x_1$-$x_{16}$ represent the 16 coefficients of a second-order truncated Chebyshev polynomial in two dimensions, ordered as follows: $1$, $x$, $x^2$, $x^3$, $y$, $xy$, $x^2y$, $x^3y$, $y^2$, $xy^2$, $x^2y^2$, $x^3y^2$, $y^3$, $xy^3$, $x^2y^3$, and $x^3y^2$.

- Darcy flow problem, AMG preconditioning:

$$1.0 + \frac{1.0}{1.0 + \frac{1.0}{x_{16} + x_3 + x_9^2 + 2.0}}$$

  Parameter meanings: $x_1$-$x_{16}$ represent the 16 coefficients as described above.

- Biharmonic Equation, SOR preconditioning:

$$\left(\frac{1.0x_2}{-4x_2 - 1.0 + \frac{1.0}{x_2}} + 1.0\right)^4$$

  Parameter meanings: $x_1$ and $x_2$ represent the length and width of the equation's boundary, respectively.

- Biharmonic Equation, AMG preconditioning:

$$1.0 + \frac{1.0}{3.0 + \frac{1.0x_1 + 1.0}{x_2}} + \frac{1.0}{x_2}$$

  Parameter meanings: $x_1$ and $x_2$ represent the length and width of the equation's boundary, respectively.

- Poisson Equation, SOR preconditioning:

$$\sqrt{\exp\left(\frac{1.0}{x_3 + \exp(2\exp(x_1^2))}\right)}$$

Parameter meanings: $x_1$-$x_8$ represent the coefficients of two second-order truncated Chebyshev polynomials for the boundary functions.

- Poisson Equation, SSOR preconditioning:

$$1.15024107160485\left(0.106506978919201x_2^2 + 1\right)^{1/16}$$

Parameter meanings: $x_1$-$x_8$ represent the coefficients of two second-order truncated Chebyshev polynomials for the boundary functions.

- Poisson Equation, AMG preconditioning:

$$\frac{1.0}{x_8^2 + 7.0}$$

Parameter meanings: $x_1$-$x_8$ represent the coefficients of two second-order truncated Chebyshev polynomials for the boundary functions.

- Thermal problem, SOR preconditioning:

$$\exp\left(\frac{0.778800783071405}{(1 - x_6)^{1/4}}\right)^{1/4}$$

Parameter meanings: $x_1$-$x_4$ represent the coefficients of the Chebyshev polynomial for the boundary temperature function on the upper and lower boundaries, while $x_5$ and $x_6$ represent the coefficients for the boundary temperature function on the left and right boundaries.

- Thermal problem, SSOR preconditioning:

$$1.0 - \frac{1.0}{\log\left(4.0(1 - 0.5x_6)^2\right)}$$

Parameter meanings: $x_1$-$x_4$ represent the coefficients of the Chebyshev polynomial for the boundary temperature function on the upper and lower boundaries, while $x_5$ and $x_6$ represent the coefficients for the boundary temperature function on the left and right boundaries.

- Thermal problem, AMG preconditioning:

$$\frac{1.0}{2.71828182845905\exp(0.135335283236613x_6) + 8.15484548537714}$$

Parameter meanings: $x_1$-$x_4$ represent the coefficients of the Chebyshev polynomial for the boundary temperature function on the upper and lower boundaries, while $x_5$ and $x_6$ represent the coefficients for the boundary temperature function on the left and right boundaries.

## E.2 Multi-Parameter Experiments

To further validate the ability of SymMaP to predict multiple parameters, we conducted experiments on AMG preconditioning with an SOR smoother, optimizing two parameters simultaneously: AMG's threshold parameter $\theta$ (default in PETSc: 0) and the SOR relaxation factor $\omega$.

All other settings match the AMG preconditioning described in the main experiment. We jointly optimize both parameters, using the condition number as the evaluation metric.

Table 8: Condition number comparison across different PDE problems and parameter settings.

| Problem | Matrix Size | None | $\theta = 0$ | | | $\theta = 0.2$ | | | $\theta = 0.8$ | | | Optimal Constant | SymMaP |
|---|---|---|---|---|---|---|---|---|---|---|---|---|---|
| | | | $\omega = 0.1$ | $\omega = 1$ | $\omega = 1.9$ | $\omega = 0.1$ | $\omega = 1$ | $\omega = 1.9$ | $\omega = 0.1$ | $\omega = 1$ | $\omega = 1.9$ | | |
| Darcy Flow | 10,000 | 752862 | 8032 | 7623 | 8321 | 21464 | 17326 | 18354 | 12496 | 9341 | 9852 | 6432 | **4021** |
| Elliptic PDE | 40,000 | 6792 | 168.2 | 161.2 | 163.8 | 191.5 | 183.2 | 187.3 | 213.4 | 194.2 | 198.4 | 159.3 | **156.1** |
| Poisson | 10,000 | 1242 | 4.31 | 4.71 | 3.56 | 67.43 | 70.34 | 54.26 | 67.43 | 70.34 | 54.26 | 3.45 | **2.94** |
| Thermal | 2,800 | 7325 | 11.2 | 10.3 | 12.6 | 463.2 | 432.1 | 489.2 | 463.2 | 432.1 | 489.2 | 8.64 | **7.53** |
| Boltzmann | 50,000 | 9548 | 246.4 | 195.7 | 197.3 | 263.9 | 213.4 | 219.3 | 353.5 | 246.3 | 283.9 | 189.2 | **165.3** |
| Lippmann-Schwinger | 50,000 | 45257 | 4832 | 4681 | 4937 | 12842 | 9853 | 9475 | 6583 | 5834 | 5931 | 4372 | **2395** |

This Table 8 demonstrates SymMaP's ability to handle multiple preconditioner parameters simultaneously.

## E.3 Custom Metric Experiments

In the main paper, our experimental focus was primarily on SOR, SSOR, and AMG preconditioners. We selected these methods because they exhibit a non-monotonic relationship between their key parameters (e.g., the relaxation factor $\omega$ or the threshold parameter $\theta_T$) and performance metrics such as solution time or condition number. This characteristic creates a non-trivial optimization landscape with a distinct optimal parameter, making these preconditioners ideal for demonstrating the core capabilities of our symbolic discovery framework, SymMaP.

We acknowledge that other important preconditioners, such as Incomplete Cholesky (ICC) and Incomplete LU (ILU) factorization, present a different type of optimization challenge. For these methods, parameters like the fill-in level typically have a monotonic relationship with individual performance metrics. For instance, increasing the fill-in level generally decreases the condition number of the preconditioned matrix but monotonically increases the computational cost of the factorization. A practical application, therefore, requires balancing this trade-off by defining a custom, often heuristic, objective function. To maintain clarity and avoid introducing ambiguity from such custom metrics in our main experiments, we initially prioritized the more straightforward optimization cases offered by SOR, SSOR, and AMG.

To demonstrate the broader applicability and generalizability of SymMaP, we conducted an additional experiment on ICC preconditioning. In this scenario, the goal is to find an optimal fill-in level that balances the trade-off between matrix conditioning and computational cost. We defined a hybrid objective function to capture this balance:

$$\text{Objective} = 0.03 \times \text{Condition Number} + \text{Time(s)} \tag{20}$$

The experiment was performed on the symmetric second-order elliptic PDE problem (with a matrix size of 40,000), maintaining consistency with the experimental settings described in the main paper. The task for SymMaP was to discover a symbolic expression for the optimal fill-in level that minimizes the objective function defined in Equation 20.

Table 9: Performance comparison for ICC preconditioning on the second-order elliptic PDE dataset. The objective value is calculated using Equation 20, where a lower value is better.

| Method | None | Default | Optimal Constant | SymMaP |
|---|---|---|---|---|
| Objective Value | 227.66 | 24.01 | 22.74 | **22.27** |

The results, presented in Table 9, show that SymMaP successfully identifies a parameterization that outperforms the default and optimal constant strategies, achieving the lowest objective value. This experiment confirms that SymMaP is not limited to preconditioners with non-monotonic optimization

landscapes but can also effectively handle trade-off-based optimization problems by learning symbolic policies for custom, multi-objective metrics. This demonstrates the framework's flexibility and generalizability to a wider range of preconditioning scenarios.

### E.4 Generalizability Analysis

#### E.4.1 Generalization Across PDE Parameters

If the matrix properties remain stable under parameter variation, the learned symbolic expressions can generalize beyond the training range. Otherwise, as noted, performance may degrade. To validate this, we conducted additional experiments:

Second-order elliptic equation, grid size: 40,000, test range: $\alpha \in [-2, 2]$, coupling term $\in [-0.5, 0.5]$ (vs. training range: $\alpha \in [-1, 1]$, coupling term $\in [-0.01, 0.01]$), tolerance: $10^{-3}$, preconditioner: SOR, linear solver: GMRES.

Table 10: Comparison of average computation times (seconds) for SOR with different $\omega$ selections, and tolerance is 1e-3.

|          | None  | PETSc default 1 | Fixed constant 0.1 | Fixed constant 1.9 | Optimal constant | SymMaP |
|----------|-------|-----------------|--------------------|--------------------|------------------|--------|
| Time (s) | 16.98 | 4.04            | 1.26               | 15.2               | 0.94             | **0.86** |

The results in Table 10 demonstrate that the expressions learned by SymMaP perform well even outside the range of training parameters, indicating that SymMaP has strong generalization capabilities.

#### E.4.2 Generalization Across Matrix Size

In preliminary tests, we observed that for certain preconditioners (e.g., SOR), the optimal preconditioning parameters are largely independent of the matrix size (resolution) for a given problem. To ensure consistency, we fixed the matrix size in our experiments.

To validate the reasonableness of this observation, we conducted additional experiments. Specifically, we trained a model on a dataset composed of matrices with mixed side lengths (uniformly $1 \times 10^3$ to $5 \times 10^4$) and tested it on a test set with similarly mixed side lengths.

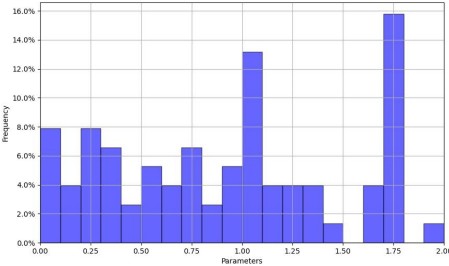

Figure 3: SOR optimal parameter distribution for a mixed edge-length second-order elliptic PDE dataset.

Table 11: SOR experiment on mixed-size datasets (uniformly $1 \times 10^3$ to $5 \times 10^4$). Other settings are consistent with the main experiment. Average time is used as the metric (in seconds).

| Dataset      | None | PETSc default $\omega = 1$ | Fixed constant $\omega$ $\omega = 0.1$ | $\omega = 0.2$ | $\omega = 1.8$ | $\omega = 1.9$ | Optimal constant | SymMaP |
|--------------|------|------|------|------|------|------|------|--------|
| Biharmonic   | 745  | 258  | 542  | 512  | 231  | 242  | 193  | **175** |
| Darcy Flow   | 164  | 65.3 | 102  | 84.1 | 51.7 | 61.8 | 47.4 | **39.5** |
| Elliptic PDE | 27.2 | 18.6 | 19.2 | 20.8 | 15.2 | 16.7 | 12.4 | **10.8** |
| Poisson      | 16.7 | 10.3 | 11.5 | 10.8 | 10.2 | 12.9 | 8.53 | **8.12** |
| Thermal      | 87.3 | 26.4 | 83.4 | 80.2 | 46.2 | 48.5 | 24.5 | **22.3** |

The distribution of the optimal parameters across the second-order elliptic PDE dataset is shown in Figure 3. The results in Table 11 confirm that the matrix size has minimal impact on the optimal parameters for SOR. Furthermore, for preconditioners where the matrix size does influence the parameters, SymMaP can seamlessly incorporate the size as an additional input feature.

### E.4.3 Generalization Across Variable Geometries

We conducted a experiment where the problem's complexity arises from a varying geometric domain rather than from PDE coefficients. We chose a second-order elliptic PDE defined on a variable triangular domain. For each problem instance, the triangle's three vertex coordinates were randomly sampled. These six coordinates were then provided as input features to SymMaP, which was tasked with predicting the optimal relaxation factor, $\omega$, for the SOR preconditioner. The performance was measured by the total solution time.

Table 12: Comparison of average computation times (in seconds) for SOR preconditioning on a second-order elliptic PDE with a variable triangular domain.

| Method | None | Default ($\omega = 1$) | $\omega = 0.2$ | $\omega = 1.8$ | Optimal Constant | Theoretical Optimal | SymMaP |
|---|---|---|---|---|---|---|---|
| Time (s) | 16.7 | 8.3 | 9.04 | 9.01 | 7.98 | 7.65 | 7.72 |

The results, shown in Table 12, indicate that SymMaP performs robustly, achieving a solution time very close to that of the empirically determined optimal constant. This experiment validates that SymMaP can effectively learn policies for problems with geometric variations by parameterizing the domain shape and using those parameters as input features.

### E.5 Error Reduction Trajectory

We evaluated the performance of different parameter settings on the biharmonic equation dataset under varying tolerance settings. Specifically, we measured the solution time and plotted the error reduction curves. Other settings are consistent with the main experiment. As shown in Figure 4, the experimental results demonstrate that SymMaP consistently delivers strong performance across all accuracy levels.

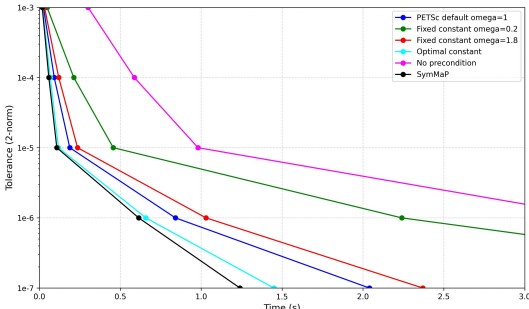

Figure 4: Error reduction curves for different parameter settings on the biharmonic equation dataset.

### E.6 Statistical Insights from the Second-Order Elliptic Equation Experiments

Here, we use the second-order elliptic equation as an example to present more details of the statistical characteristics of experimental data, as shown in Table 13.

Table 13: Performance comparison for solving a second-order elliptic equation with an edge length of 40,000 and a tolerance of $10^{-7}$ using SOR preconditioning. The coefficients range from $(-0.1, 0.1)$ for the main parameters and $(0.005, 0.01)$ for the coupling terms. Average time is used as the metric (in seconds).

| Statistic | None | PETSc default 1 | Fixed constant 0.1 | Fixed constant 1.9 | Fixed constant | SymMaP |
|---|---|---|---|---|---|---|
| Mean | 48.10 | 33.02 | 36.54 | 28.09 | 26.28 | **22.67** |
| Median | 11.79 | 9.87 | 9.99 | 8.53 | 8.42 | **8.04** |
| Upper quartile | 63.67 | 42.81 | 49.54 | 39.78 | 35.79 | **29.67** |
| Lower quartile | 3.38 | 2.68 | 2.65 | 2.31 | 2.10 | **2.10** |
| Variance | 1365.1 | 822.47 | 864.45 | 784.92 | 749.35 | **705.27** |

### E.7 Comparative Experiments with Learning-Based Approaches

SymMaP enhances traditional preconditioning algorithms by discovering symbolic expressions for optimal parameters. In our experimental design, we deliberately avoid direct comparisons with other learning-based approaches for the following reasons:

1. Focus on Preconditioning Enhancement: SymMaP is specifically designed to improve traditional preconditioners without altering their core algorithms. A fair comparison would require other learning-based methods that also preserve the original preconditioning process. However, no such methods currently exist.

2. CPU Compatibility: Linear solver environments are predominantly CPU-based. SymMaP's symbolic expressions integrate seamlessly into CPU workflows, whereas neural network-based approaches often lack efficient CPU deployment.

3. Generality: SymMaP is highly adaptable to diverse preconditioners and linear solvers. In contrast, existing learning-based methods either: (a) combine with specific preconditioners (e.g., [41], [21]), or (b) target narrow use cases (e.g., [60], [23] for ICC-preconditioned CG; [34] exclusively for CG).

4. Interpretability & Safety: Numerical algorithms demand rigorous analysis. Opaque predictors, such as neural networks, risk violating theoretical constraints (e.g., avoiding $\omega \approx 0$ or 2 in SOR). SymMaP's symbolic expressions enable proactive avoidance of such issues through analytical guarantees.

While these advantages are significant, we also conducted comparative experiments with other learning-based preconditioning algorithms to evaluate their performance. The experiments are divided into two groups: 1. Table 14 focuses on symmetric matrices, using the CG algorithm with a tolerance of $10^{-7}$. 2. Table 15 targets nonsymmetric matrices, employing the GMRES algorithm with a tolerance of $10^{-7}$.

All other settings are consistent with those described in the paper. The performance metric is the average solve time, measured in seconds, where lower values indicate better performance.

Table 14: Comparison of average computation times (seconds) with different methods, linear solver is CG, and tolerance is 1e-7.

| Dataset | Matrix Size | Configuration | None | SSOR | AMG | AMG+SSOR | ILU | ICC | [41] | [41]+SSOR | [41]+ICC | [41]+AMG | [21] | [21]+SSOR | [34] | [41]+[34] | [21]+[34] | [23] | [41]+[23] | [21]+[23] | [60] | [41]+[60] | [21]+[60] | [24] | SymMaP(SSOR) | [41]+SymMaP(SSOR) | [21]+SymMaP(SSOR) |
|---|---|---|---|---|---|---|---|---|---|---|---|---|---|---|---|---|---|---|---|---|---|---|---|---|---|---|---|
| Darcy flow | 4900 | CPU & GPU | 4.18 | 0.49 | 0.55 | 0.45 | 0.45 | 0.62 | 2.09 | 0.31 | 0.61 | 0.62 | 0.47 | 0.42 | 0.52 | 0.51 | × | 0.49 | 0.49 | × | 0.47 | 0.46 | × | × | 0.41 | 0.28 | 0.38 |
| | | Only CPU | 4.18 | 0.49 | 0.55 | 0.45 | 0.45 | 0.62 | 2.14 | 0.35 | 0.66 | 0.66 | 0.50 | 0.45 | 0.55 | 0.57 | × | 0.56 | 0.53 | × | 0.49 | 0.53 | × | × | 0.41 | 0.32 | 0.44 |
| Elliptic PDE | 40000 | CPU & GPU | 23.9 | 10.5 | 15.7 | 9.4 | 9.4 | 18.6 | 11.2 | 7.5 | 20.3 | 15.5 | 14.2 | 9.3 | 8.6 | 8.9 | × | 8.4 | 8.7 | × | 8.4 | 8.7 | × | × | 7.7 | 5.5 | 7.2 |
| | | Only CPU | 23.9 | 10.5 | 15.7 | 9.4 | 9.4 | 18.6 | 11.5 | 7.8 | 20.6 | 15.8 | 14.3 | 9.4 | 8.7 | 9.2 | × | 8.5 | 9.0 | × | 8.4 | 9.1 | × | × | 7.7 | 5.8 | 7.3 |
| Lippmann-Schwinger | 50000 | CPU & GPU | 60.3 | 30.5 | 35.2 | 28.4 | 28.4 | 32.5 | 30.2 | 16.4 | 31.2 | 33.2 | 33.2 | 22.7 | 28.9 | 17.2 | × | 27.3 | 27.1 | × | 27.1 | 26.2 | × | × | 25.4 | 12.5 | 18.9 |
| | | Only CPU | 60.3 | 30.5 | 35.2 | 28.4 | 28.4 | 32.5 | 30.5 | 16.7 | 31.5 | 33.5 | 33.3 | 22.8 | 29.1 | 17.5 | × | 27.4 | 27.2 | × | 27.2 | 26.7 | × | × | 25.4 | 12.8 | 19.0 |

Table 15: Comparison of average computation times (seconds) with different methods, linear solver is GMRES, and tolerance is 1e-7.

| Dataset | Matrix Size | Configuration | None | SOR | AMG | AMG+SOR | ILU | [41] | [41]+SOR | [41]+ILU | [41]+AMG | [21] | [21]+SOR | [34] | [41]+[34] | [21]+[34] | [23] | [41]+[23] | [21]+[23] | [60] | [41]+[60] | [21]+[60] | [24] | SymMaP(SOR) | [41]+SymMaP(SOR) | [21]+SymMaP(SOR) |
|---|---|---|---|---|---|---|---|---|---|---|---|---|---|---|---|---|---|---|---|---|---|---|---|---|---|---|
| Biharmonic | 4200 | CPU & GPU | 7.67 | 2.04 | 4.21 | 1.93 | 2.01 | 4.31 | 1.64 | 1.94 | 4.18 | 4.12 | 1.82 | × | × | × | × | × | × | × | × | × | × | 1.24 | 1.01 | 1.21 |
| | | Only CPU | 7.67 | 2.04 | 4.21 | 1.93 | 2.01 | 4.35 | 1.69 | 1.99 | 4.24 | 4.14 | 1.84 | × | × | × | × | × | × | × | × | × | × | 1.24 | 1.05 | 1.23 |
| Elliptic PDE | 40000 | CPU & GPU | 31.3 | 21 | 24.3 | 18.7 | 23.5 | 22.4 | 13.2 | 22.1 | 24.4 | 23.4 | 17.3 | × | × | × | × | × | × | × | × | × | × | 15.8 | 10.6 | 14.3 |
| | | Only CPU | 31.3 | 21 | 24.3 | 18.7 | 23.5 | 22.6 | 13.4 | 22.4 | 24.7 | 23.5 | 17.4 | × | × | × | × | × | × | × | × | × | × | 15.8 | 10.9 | 14.4 |
| Boltzmann | 50000 | CPU & GPU | 32.1 | 12.4 | 18.9 | 10.2 | 13.2 | 15.2 | 9.8 | 12.9 | 19.2 | 17.2 | 9.8 | × | × | × | × | × | × | × | × | × | × | 10.2 | 7.2 | 8.1 |
| | | Only CPU | 32.1 | 12.4 | 18.9 | 10.2 | 13.2 | 15.5 | 10.2 | 13.2 | 19.5 | 17.3 | 10 | × | × | × | × | × | × | × | × | × | × | 10.2 | 7.4 | 8.2 |

### E.7.1 Notes on These Methods

**Classification of Existing Algorithms**:

1. Methods Enhancing Linear Solvers: These approaches add optimization modules or modify the iteration process (e.g., [41], [21]). These methods target different components than SymMaP and could potentially be combined with our approach for further improvements.

2. Methods Predicting Preconditioning Matrices: These approaches use neural networks to predict preconditioning matrices (e.g., [60], [34], [24]).

**Code Availability**:

- [34]: The official implementation is available on GitHub (NeuralPCG).
- [23]: The official implementation is available on GitHub (NeuralIF).
- [60]: The official implementation is available on OpenReview (PreCorrector).
- [41]: The official implementation is available on GitHub (NeurKItt).
- [21]: The official implementation is available on GitHub (Learning to Optimize Multigrid Solvers).
- [24]: The official implementation is available on GitHub (Learning Neural PDE Solvers with Convergence Guarantees).

**Table details explanation**:

- Combinations: Sequential application of methods (e.g., 'AMG+?' or '[41]+?').
- ×: Method incompatible (e.g., [34], [23], [24], [60] only work for symmetric matrices/CG) or excessively slow (e.g., [24] >100s for Elliptic PDE).
- None: No preconditioning.
- SOR/SSOR/AMG/ICC/ILU: PETSc defaults.
- SymMaP(SOR/SSOR): Optimized via SymMaP.
- [41]: Accelerates Krylov solvers via recycling (distinct from preconditioning); combinable with other methods.
- [21]: An AMG variant; allows nested preconditioning but conflicts with learning-based methods ([34], [23], [60]).
- [24]: A fixed-point iteration (non-Krylov); computationally prohibitive for large sparse systems.

**Experimental Setup**:

- Hardware:
    - CPU: AMD Ryzen 9 5900HX, MPI 8-core parallel.
    - GPU: RTX 3090 (CPU-only uses PyTorch CPU inference).
    - Averaging: 1000 runs per problem.
- Linear Solver: For fairness, we use PETSc's CG/GMRES.

### E.7.2 Results

As shown in Tables 14 and 15, across all experiments, the combination of "[41]+SymMaP" consistently achieved the shortest computation times. When evaluating standalone algorithms (without combinations), SymMaP alone demonstrated the fastest performance in all test cases. The performance advantage of SymMaP was particularly pronounced in CPU-only environments. SymMaP generalizes to any parameterized preconditioner, unlike other methods (e.g., [23] for ICC, [21] for AMG). These results conclusively demonstrate the superior performance characteristics of the SymMaP approach.

## E.8 Interpretable Analysis Details

Table 16: Symbolic expressions learned from the main experiments

| Precondition | Dataset | Symbolic expression |
|---|---|---|
| SOR | Biharmonic | $1.0 + 1.0/(4.0 + 1.0/x_2) + 1.0/x_1$ |
| SOR | Elliptic PDE | $1.0 + 1.0/(x_2 + 1.0 + 1.0/(x_2 + 4.0))$ |
| SOR | Darcy Flow | $1.0 + 1.0/(x_4 + 1.0)$ |
| SSOR | Elliptic PDE | $1.0 + 1.0/(x_2 + 1.2)$ |
| AMG | Elliptic PDE | $(x_1 x_3 + 1)/7$ |

As shown in Table 16, the variables are defined as follows: in the first row, $x_1$ and $x_2$ represent the size of the boundary for PDE solutions; in the second row, $x_2$ represents the coefficient of a second-order coupling term; in the third row, $x_4$ is the coefficient of the fourth x-term multiplied by the first y-term in a two-dimensional Chebyshev polynomial; in the fourth row, $x_2$ again denotes the coefficient of a second-order coupling term; in the fifth row, $x_1 x_3$ signifies the coefficient of a second-order non-coupling term.

## E.9 Analysis of Hyperparameters

The performance of SymMaP is primarily influenced by the learning rate of the RNN, batch size, and dataset size. We conducted experiments to study the impact of these hyperparameters.

**Symbolic Learning RNN Parameters**:

Table 17: Performance comparison of SymMaP under various symbolic learning RNN parameters (lower condition numbers are preferable). The experiment focuses on optimizing AMG preconditioning coefficients in the Darcy Flow dataset.

| Learning Rate | Batch Size | Condition number | Training time(s) |
|---|---|---|---|
| | 500 | 6780 | 1173.09 |
| 0.01 | 1000 | 5168 | 863.51 |
| | 2000 | 6898 | 522.80 |
| | 500 | 5935 | 1104.16 |
| 0.001 | 1000 | 11774 | 676.40 |
| | 2000 | 5935 | 505.85 |
| | 500 | 4718 | 1026.45 |
| 0.0005 | 1000 | 5935 | 703.17 |
| | 2000 | 5935 | 549.36 |
| | 500 | 12228 | 1324.00 |
| 0.0001 | 1000 | 7508 | 837.18 |
| | 2000 | 6884 | 603.62 |

Results in Table 17 indicate that an appropriate combination of RNN learning rate and batch size can enhance performance.

**Dataset size**:

Table 18: Performance comparison of SymMaP across varying dataset sizes (lower condition numbers indicate better performance). The experiment evaluates the optimization of AMG preconditioning coefficients for the Darcy Flow dataset.

| Dataset size | Condition number | Training time (s) |
|---|---|---|
| 10 | 7032 | 669.68 |
| 50 | 6980 | 737.80 |
| 100 | 4892 | 812.02 |
| 500 | 3811 | 699.54 |
| 1000 | 5345 | 703.17 |

Table 18 demonstrates that increasing the dataset size enhances the performance of symbolic expressions learned by SymMaP, as expected.

## E.10 Analysis of Data Generation Method

The SymMaP framework is modular, allowing the data generation step to be performed by any suitable search algorithm. We conducted this comparison on the second-order elliptic PDE problem with the SOR preconditioner (matrix size 40,000, 1,000 instances), evaluating standard grid search, random search, and Bayesian optimization against our adaptive approach. Table 19 summarizes the results, showing each method's data generation time, the Mean Absolute Error (MAE) of the discovered optimal parameters against our high-precision reference, and the final solution time achieved by SymMaP when trained on the resulting dataset.

Table 19: Comparison of different data generation methods for finding the optimal SOR parameter.

| Data Generation Method | Avg. Generation Time | MAE (vs. SymMaP) | Final Avg. Solution Time (s) |
|---|---|---|---|
| Standard Grid Search | $\sim$100h | $7.80 \times 10^{-4}$ | 16.0 |
| Random Search (2000 samples) | $\sim$100h | $1.74 \times 10^{-3}$ | 16.0 |
| Bayesian Optimization | $\sim$11h | $9.80 \times 10^{-4}$ | 15.9 |
| Adaptive Search (SymMaP) | $\sim$25h | 0 | 15.9 |

The analysis reveals two key findings. First, while methods like Bayesian optimization are significantly more time-efficient for data generation, all tested algorithms converge to nearly identical optimal parameters (MAE $< 1.74 \times 10^{-3}$). Second, because the resulting training data is consistent, SymMaP discovers the same symbolic expression and achieves the same final performance regardless of the data source. This demonstrates that SymMaP's effectiveness is robust and independent of the specific search algorithm used, provided it accurately identifies the optimal values.

## E.11 Comparison with Alternative Interpretable and Symbolic Methods

To clarify the advantages of SymMaP, we conducted additional experiments comparing it against other lightweight, interpretable, and symbolic modeling approaches. We categorize these alternatives into two groups: (1) standard interpretable models like linear and polynomial regression, and (2) alternative symbolic regression paradigms, such as Genetic Programming (GP), that do not use an RNN-based search.

### E.11.1 Comparison with Standard Interpretable Models

A key advantage of symbolic regression over fixed-form models is its flexibility in both feature selection and functional form. While models like linear or polynomial regression are interpretable, their expressive power is limited. In contrast, SymMaP explores a much richer space of mathematical expressions. In fact, by constraining the library of operators (e.g., to only '+' and '*'), our framework can effectively subsume these simpler models, which represent special cases within the broader search space.

To provide a direct performance comparison, we conducted an experiment on the second-order elliptic PDE problem (using the SOR preconditioner, with a matrix size of 40,000 and 1,000 data points). We tasked Linear Regression, Quadratic Polynomial Regression, and SymMaP with predicting the optimal SOR relaxation factor.

Table 20: Performance comparison against standard interpretable models.

| Fitting Method | Solution Time (s) |
| --- | --- |
| Linear Regression | 20.9 |
| Quadratic Polynomial Regression | 17.0 |
| SymMaP | **15.8** |

As shown in Table 20, SymMaP discovers a policy that yields significantly faster solution times. This demonstrates that its expressive power is crucial for accurately capturing the complex, non-linear landscape of optimal preconditioning parameters.

### E.11.2 Comparison with Genetic Programming-Based Symbolic Regression

A more direct comparison is against other symbolic regression paradigms. Our choice of an RNN+RL approach was primarily motivated by its superior scalability to the high-dimensional feature spaces often encountered in preconditioning problems. For instance, the Darcy Flow dataset in our paper has 16 input features. Genetic Programming (GP) algorithms, which form the basis of popular tools like PySR, are known to scale poorly with a large number of features. PySR itself warns users that its performance can degrade in high-dimensional settings, recommending dimensionality reduction as a prerequisite.

To quantify this, we conducted a head-to-head comparison between SymMaP and PySR on finding the optimal SOR relaxation factor for the Darcy Flow and Biharmonic datasets. For PySR, we report the result from the best-performing expression found across five runs.

Table 21: Performance comparison of SymMaP against PySR.

| Dataset | None | Default | Optimal Constant | PySR | SymMaP |
| --- | --- | --- | --- | --- | --- |
| Darcy Flow | 33.1 | 13.5 | 9.54 | 8.87 | **8.50** |
| Biharmonic | 7.67 | 2.04 | 1.31 | 1.41 | **1.24** |

The results in Table 21 show that while both methods find effective expressions, SymMaP consistently outperforms PySR. Notably, for the Biharmonic problem, the expression found by PySR failed to improve upon the optimal fixed constant, likely due to the performance limitations of GP in this context. Furthermore, the RNN+RL approach was significantly more efficient, requiring approximately 15 minutes to discover its expressions, whereas PySR took over 1 hour.

In summary, for the specific challenge of preconditioning parameter prediction, which often involves a larger number of input features, the RNN+RL search strategy in SymMaP offers superior performance and greater efficiency compared to both simpler interpretable models and alternative GP-based symbolic regression tools.

