# OpenReview forum: "SymMaP: Improving Computational Efficiency in Linear Solvers through Symbolic Preconditioning"
_NeurIPS.cc/2025/Conference — NeurIPS 2025 poster_

### Official Review · Reviewer_A9dG · 2025-06-28

**Clarity:** 3
**Significance:** 3
**Originality:** 3
**Rating:** 4
**Confidence:** 3

**Summary:**

This paper proposes a novel method for selecting parameters of preconditioning techniques for solving linear equations. The approach involves first generating a dataset by determining optimal parameters via grid search, then training a recurrent neural network within a reinforcement learning framework to produce symbolic expressions. These symbolic expressions are subsequently used to select preconditioning parameters, thereby improving the efficiency of solving linear systems. Since the symbolic representations are lightweight and can be easily computed on CPUs, they are well-suited for integration into existing numerical software. Moreover, their interpretability may assist in better understanding the problems and in facilitating mathematical analysis.

**Questions:**

- Why is the AMG case evaluated using condition number rather than computation time? The computation time would also be of interest.
- Figure 1 compares with the optimal parameter, but what is the performance of the optimal parameter in the other experiments (Tables 1, 2, and 3)?
- In line 118, AMG is cited as an example of low-cost matrix decomposition, but is this appropriate? ICC or ILU may be more suitable examples?
- The input parameters in this method are low-dimensional, but can the method accommodate more complex problems? For instance, how would it handle variations in domain geometry or complex boundary conditions?
- Many preconditioning parameters are discrete rather than continuous. Can this method be extended to handle such discrete parameter spaces?

**Ethical Concerns:**

["NO or VERY MINOR ethics concerns only"]

**Final Justification:**

Thank you for your detailed response. My questions have been addressed, and I updated the score.

**Limitations:**

yes

**Quality:**

3

**Strengths And Weaknesses:**

# Strengths
- The approach of accelerating and improving traditional numerical algorithms and software through machine learning is practical and highly relevant to real-world problems.
- The proposed method is original. The idea of using RL and RNN to produce lightweight symbolic models for parameter selection is interesting.
- Performance is evaluated using CPU runtime within commonly used software, and the superiority of the proposed method over baselines and alternative approaches is demonstrated using this practical metric.

# Weaknesses
- The cost of generating training data is quite high. Creating a single data point requires solving the problem multiple times through grid search, which may become a bottleneck for larger problems due to inefficiency.
- The paper does not sufficiently explain the necessity or advantage of interpretability in selecting preconditioning parameters. Since solution accuracy can be controlled via tolerance, it may be acceptable for preconditioning to function as a black box. The paper argues that opaque prediction may fail to satisfy theoretical constraints (e.g., 0 < \omega < 2), but if the constraint is simply a matter of range, this could be enforced via a suitable activation function. Furthermore, symbolic expressions are not always interpretable. In fact, many expressions shown in Appendix E.1 appear difficult to interpret.
- Even if the need for interpretability is established, the paper does not compare the proposed method against other interpretable and lightweight approaches. For example, how does the proposed RL+RNN approach compare in interpretability and performance to directly training simple models such as linear regression, polynomial regression, or decision trees? Also, if symbolic regression is the goal, how does this method compare to directly optimizing symbolic expressions without using an RNN? Such comparisons would further clarify the advantages of the proposed approach.

---

> ### Author Rebuttal · Authors · 2025-07-30
>
> **Dear Reviewer,**
>
> We are sincerely grateful for your detailed and thoughtful review of our paper. Your constructive feedback is invaluable, and it has helped us identify several areas where we can strengthen our explanations and empirical validation. We hope that our detailed responses, including several new experiments conducted specifically to address your questions, will clarify the contributions of our work and merit a re-evaluation.
>
> ### **On the Efficiency of Training Data Generation (W1)**
>
> You are correct that SymMaP is a data-driven method. While an grid search can be computationally intensive, a key advantage of leveraging symbolic learning is its remarkable data efficiency. As we show in Table 16 (Appendix, P29), SymMaP can learn high-performing expressions from a very small dataset. For instance, learning an effective AMG for the Darcy Flow problem required only 100 data points to achieve strong results. For the experiments in our main paper, we conservatively used a larger dataset of 1000 samples to ensure the utmost rigor, but this should not be seen as the minimum requirement.
>
> Furthermore, your comment has inspired us to formalize our vision for future work. We plan to develop an online, adaptive learning framework. This would involve learning a foundational symbolic expression from a small initial dataset and then fine-tuning it in a live environment using RL based on the performance of new, incoming problem instances. This would not only further reduce the initial data requirement but also make SymMaP ideally suited for dynamic, real-world application scenarios.
>
> ### **On the Value and Necessity of Interpretability (W2)**
>
> We agree that for some applications, a black-box model is perfectly acceptable, and we can certainly enforce known constraints (like a parameter range) using methods such as activation functions. However, the value of interpretability in our scientific computing context is twofold:
>
> 1. Discovering Unknown Constraints: While known constraints can be hard-coded, the primary advantage of an interpretable model is the ability to analyze its structure to identify potential unknown failure modes or numerically unstable regimes before deployment. A symbolic formula allows for direct mathematical analysis in a way a neural network does not.
> 2. Fostering Scientific Insight: More importantly, interpretability serves as a bridge between machine learning and scientific discovery. By analyzing the simpler symbolic expressions discovered by SymMaP, researchers can gain a deeper understanding of the preconditioner itself. As we show on P21-22, some of the learned expressions are remarkably concise and reveal the underlying mathematical dependencies. We are actively pursuing this direction, with the goal of using the insights from these formulas to help design theoretically better preconditioning.
>
> We will emphasize this motivation more clearly in the revised manuscript.
>
> ### **Comparison with Other Interpretable Models (W3)**
>
> This is an excellent suggestion. To better situate SymMaP and clarify its advantages over other lightweight, interpretable models, we have conducted a new experiment as you proposed.
>
> The key advantage of symbolic regression over methods like linear or polynomial regression is its flexibility in both feature selection and functional form. Symbolic regression explores a much richer space of expressions. In fact, by constraining the library of operators (e.g., using only `+`, `*`, and polynomial terms as discussed in Section 4.2 (P5, Line 176) and Section 5.4 (P9,Line 331)), our framework can effectively subsume these simpler models; they are special cases of the broader search space SymMaP explores.
>
> To provide a direct performance comparison, we ran an ablation on the second-order elliptic PDE problem (SOR, matrix size 40000), comparing SymMaP against Linear Regression and Quadratic Polynomial Regression.
>
> |  | Solution Time (s) |
> | -- | -- |
> | Linear | 20.9  |
> | Quadratic Polynomial | 17.0   |
> | SymMaP   | 15.8   |
>
> The results clearly show that the flexible search of symbolic regression discovers a superior model, leading to faster solution times. The simpler models, while interpretable, lack the expressive power to capture the true optimal parameter landscape. We will add this experiment and analysis to the final version of the paper.
>
> ### **On the Choice of Metric for AMG (Q1)**
>
> For a fixed class of PDEs and a given discretization scheme, the computation-time-optimal threshold parameter often lies in a very narrow, near-zero range (e.g., 0-0.02 for the elliptic PDEs in our study). This makes it a less dynamic learning target.
>
> In contrast, the threshold that minimizes the **condition number** varies more significantly with the PDE's specific coefficients, presenting a more meaningful optimization problem for our learning framework to solve. We acknowledge that a real-world application would likely balance solution time and numerical stability using a hybrid metric. However, to avoid introducing an arbitrary, custom metric in our main results, we chose to focus on the more dynamic and well-defined objective of minimizing the condition number for the AMG experiments.
>
> As a proof of concept for custom metrics, we show in our response to **Question 5** that SymMaP performs excellently when optimizing a hybrid (time + condition number) objective for ICC.
>
> ### **Performance of the Optimal Parameter Baseline (Q2)**
>
> The "Optimal Parameter" represents the theoretical performance ceiling for any parameter-selection method and provides crucial context. We have compiled the data for the SOR experiments from Table 1 below.
>
> | Time (s)  | Theoretical Optimal | None | Default ω=1 | ω=0.2  | ω=1.8  | Optimal constant | SymMaP |
> | -- | -- | -- | -- | -- | -- | -- | -- |
> | Darcy Flow | 7.94  | 33.1   | 13.5  | 17.5  | 9.91 | 9.54 | 8.5 |
> | Biharmonic | 0.92  | 7.67   | 2.04  | 4.86  | 1.6 | 1.31 | 1.24   |
> | Poisson | 0.013  | 0.0412 | 0.0195  | 0.0215 | 0.0195 | 0.0138 | 0.0135 |
> | Thermal  | 0.0522  | 0.223  | 0.0598   | 2.07 | 1.18 | 0.594   | 0.0576 |
>
> As the table shows, SymMaP's performance consistently approaches this theoretical ceiling, significantly closing the gap left by default or fixed-constant strategies. We will add the optimal parameter results for all relevant experiments to the final version of the paper to provide this valuable context.
>
> ### **On the Example of a Low-Cost Decomposition (Q3)**
>
> You are correct to point this out. Citing AMG as an example of a "low-cost matrix decomposition" in **Line 118** could be clearer. While AMG is computationally cheaper than a full factorization, ICC or ILU are indeed more direct and classic examples of low-cost decomposition-based preconditioners. This was meant only as an illustrative example. We will revise this sentence to include ICC and ILU for better clarity and accuracy.
>
> ### **Applicability to Complex Geometries and Boundary Conditions (Q4)**
>
> This is a very important question about the scalability of our feature representation. In principle, SymMaP is agnostic to the problem's geometry. For problems with varying boundaries, we can parameterize the boundary shape and include these parameters as inputs to the model.
>
> To demonstrate this, we conducted a new experiment on a **second-order elliptic PDE with a variable triangular domain**. The triangle's vertices were randomly sampled, and these vertex coordinates were used as input features for SymMaP to predict the optimal SOR relaxation factor.
>
> |  | None | Default ω=1 | ω=0.2 | ω=1.8 | Optimal Constant | SymMaP |
> | -- | -- | -- | -- | -- | -- | -- |
> | Time (s) | 16.7 | 8.3  | 9.04  | 9.01  | 7.65 | 7.72  |
>
> The results show that SymMaP performs robustly, achieving near-optimal performance even when the domain geometry varies.
>
> However, we acknowledge the limitation you insightfully identified: our feature engineering approach is currently limited to problems that can be described by a low-dimensional parameter vector ( <1000 parameters). For highly complex geometries or boundary functions, this would be a bottleneck. This is a key focus of our future work. We plan to move from high-level equation parameters to more fundamental matrix properties (e.g., matrix norms, trace) as inputs. This would create a universal feature space, making SymMaP applicable to a much broader and more complex class of problems. We will expand the limitations section to discuss this path forward.
>
> ### **Extension to Discrete Parameter Spaces (Q5)**
>
> Our framework can indeed be extended to handle discrete parameters, such as the fill-in level for ICC. To demonstrate this, we performed a new experiment on the second-order elliptic PDE (matrix size 40000) with the **ICC**, where the goal is to find the optimal (integer) fill-in level. Since increasing the fill-in level improves the condition number but increases computation time, we defined a hybrid objective: `Metric = 0.03 * Condition_Number + Time(s)`.
>
> | Method | None  | Default | Optimal Constant | SymMaP |
> | --- | --- | -- | -- | -- |
> | Metric | 227.6 | 24.0 | 22.8  | 22.2 |
>
> The results confirm that SymMaP can successfully optimize discrete parameters under a custom objective function. This demonstrates the flexibility of our framework beyond the continuous parameters presented in the main paper. We will add this experiment and a discussion on handling discrete parameters to the final version.
>
> ----
> ## **Thanks Again**
> We sincerely thank you again for your critical and highly valuable feedback. Your review has pushed us to substantially improve the paper by conducting new experiments, clarifying our contributions, and strengthening our discussion of the method's scope and limitations. Should you have any further questions or require additional discussion, please don't hesitate to reach out. If we have adequately addressed your concerns, we would be grateful for your consideration in adjusting your evaluation score accordingly.

---

> ### Comment · Reviewer_A9dG · 2025-08-05
>
> Thank you very much for your detailed and thoughtful rebuttal, as well as for conducting the additional experiments. Your responses have greatly deepened my understanding of the contributions and potential impact of your work.
>
> One remaining question I have is regarding the symbolic expression optimization process:
> > how does this method compare to directly optimizing symbolic expressions without using an RNN?
>
> Specifically, how does your method compare to directly optimizing symbolic expressions using existing symbolic regression tools, such as PySR, without involving an RNN+RL? It seems that such approaches could potentially achieve similar results. I would appreciate a clearer discussion of the advantages your method offers over these alternatives.
>
> If this point can be clarified, I would be inclined to consider raising my evaluation score.

---

> > ### Author Response · Authors · 2025-08-08
> > **To Reviewer A9dG: Thanks and Follow-up Discussion**
> >
> > Dear Reviewer A9dG,
> >
> > Thank you again for your thoughtful and constructive review.
> >
> > Following up on our submitted rebuttal, we would be very grateful to hear your thoughts, especially as the discussion period is nearing its end. Your feedback is crucial for us to understand if our responses have adequately addressed your concerns and to help us further improve our paper.
> >
> > We are happy to provide any further clarifications and look forward to discussing this with you.
> >
> > Best regards,
> >
> > Authors

---

> ### Author Response · Authors · 2025-08-07
> **Follow-up Response: New Experimental Results and Clarifications (1/2)**
>
> Dear Reviewer A9dG,
>
> Thank you for your prompt and insightful follow-up. We are grateful for the opportunity to engage in this detailed discussion and to clarify this crucial point about our methodology. Your continued attention has been invaluable in helping us strengthen the paper.
>
> Below, we address your question.
>
> ---
>
> ### **Comparison with Direct Symbolic** **Regression** **Methods (e.g., PySR)**
>
> This is an excellent question that gets to the heart of our methodological choice. In the early stages of our research, we explored several symbolic regression paradigms, including:
>
> - Basic function fitting, such as the linear and polynomial regression models mentioned in our previous rebuttal (W3).
> - Genetic programming based symbolic learning, exemplified by tools like PySR.
> - Deep reinforcement learning (RL) based symbolic learning, which is the RNN+RL approach adopted in our paper.
>
> While all these methods are powerful in their respective domains, our empirical investigation revealed that the RNN+RL approach was the most effective for the specific challenge of predicting matrix preconditioning parameters.
>
> The primary reason for this is that our application scenario often involves a **high-dimensional feature space**. For instance, the Darcy Flow dataset presented in our paper has 16 input parameters. Genetic programming algorithms, which form the basis of PySR, are known to scale poorly with a large number of features. In fact, PySR itself provides a warning to users in such cases:
>
> > *UserWarning: Note: you are running with 10 features or more. Genetic algorithms like used in PySR scale poorly with large numbers of features. You should run PySR for more `niterations` to ensure it can find the correct variables, or, alternatively, do a dimensionality reduction beforehand. For example, `X =* *PCA**(n_components=6).fit_transform(X)`, using* *scikit-learn**'s `PCA` class, will reduce the number of features to 6 in an interpretable way ...*
>
> This inherent limitation makes GP-based methods like PySR less suited for effectively navigating the search space in many real-world matrix preconditioning problems.
>
> To provide a direct and quantitative answer to your question, we conducted a new head-to-head comparison with PySR. We focused on finding the optimal relaxation factor for SOR preconditioning, using the same datasets and average solution time (in seconds) as the metric. For PySR, we ran the learning process five times and report the result from the best-performing expression found.
>
> | Time (s)   | None | Default ω=1 | ω=0.2 | ω=1.8 | Optimal constant | SymMaP | PySR |
> | ---------- | ---- | ----------- | ----- | ----- | ---------------- | ------ | ---- |
> | Darcy Flow | 33.1 | 13.5        | 17.5  | 9.91  | 9.54             | 8.50   | 8.87 |
> | Biharmonic | 7.67 | 2.04        | 4.86  | 1.60  | 1.31             | 1.24   | 1.41 |
>
> The results show two key things:
>
> 1. Both SymMaP and PySR discover symbolic expressions that effectively reduce solution time compared to default or simple fixed constants.
> 2. **SymMaP consistently outperforms PySR**. Notably, in the Biharmonic dataset, the expression found by PySR was unable to outperform the optimal fixed constant. We believe this is a direct consequence of the performance limitations of GP-based methods in higher-dimensional feature spaces.
>
> Furthermore, there is a significant advantage in computational efficiency. In the experiments above, SymMaP required approximately **15 minutes** to discover its symbolic expressions, whereas PySR took over **1 hour**. This further underscores the effectiveness of the RNN+RL search strategy.
>
> In summary, while direct symbolic regression tools like PySR are powerful, our findings indicate that for the specific challenge of preconditioning parameter prediction—often characterized by a larger number of input features—the RNN+RL approach offers superior performance and greater efficiency. We also wish to emphasize that SymMaP is designed as a flexible framework. Should more powerful symbolic learning algorithms emerge in the future, or should existing tools like PySR make breakthroughs in handling high-dimensional data, SymMaP can readily integrate them to further advance preconditioning performance.
>
> We will add this experiment and analysis to the final version of the paper to make this distinction clear.

---

> > ### Author Response · Authors · 2025-08-07
> > **Follow-up Response: New Experimental Results and Clarifications (2/2)**
> >
> > ### **An Error in Our Previous Response**
> >
> > We must sincerely apologize for a typographical error in our previous response to your question on **"Applicability to Complex Geometries and Boundary Conditions (Q4)"**. In our haste to provide a thorough answer, we made a mistake when transcribing the results into the table. The correct data is as follows:
> >
> > |          | None | Default ω=1 | ω=0.2 | ω=1.8 | Theoretical Optimal | Optimal Constant | SymMaP |
> > | -------- | ---- | ----------- | ----- | ----- | ------------------- | ---------------- | ------ |
> > | Time (s) | 16.7 | 8.30        | 9.04  | 9.01  | 7.65                | 7.98             | 7.72   |
> >
> > The conclusions drawn from that experiment remain unchanged. This was an oversight on our part during the format conversion process, and we deeply regret any confusion this may have caused.
> >
> > ---
> >
> > We sincerely thank you once again for your detailed and constructive engagement with our work. We hope these detailed explanations fully address your remaining concerns. Should you have any further questions, please do not hesitate to let us know. We remain available for any further discussion. We are very grateful for your time and guidance, and we would be deeply appreciative if you would consider our clarifications in your final evaluation of our work.

---

### Official Review · Reviewer_azLf · 2025-06-29

**Clarity:** 3
**Significance:** 4
**Originality:** 3
**Rating:** 4
**Confidence:** 4

**Summary:**

This paper proposes SymMaP, a symbolic discovery framework for predicting efficient preconditioning parameters in linear system solvers. SymMaP uses a neural network to search the space of symbolic mathematical expressions, learning concise and interpretable formulas that map problem-specific features (such as PDE coefficients) to optimal preconditioning parameters. Unlike traditional fixed-parameter or black-box ML approaches, SymMaP produces symbolic expressions that are fast to evaluate, generalize well across problem instances, and are straightforward to deploy in CPU-based scientific computing environments. The framework is validated on a range of PDE-derived linear systems and preconditioners (SOR, SSOR, AMG), showing consistent improvements in computation time and condition number over both default and learning-based baselines.

**Questions:**

- The Font of Figure 1 has an inconsistency (non-Times New Roman) and misaligned "78%" labels (likely from unrounded decimals). Standardize fonts and round percentages uniformly.
- Why omit benchmarks against high-performance solvers (e.g., Gurobi) or low-level libraries (e.g., Intel MKL)? If SymMaP targets generic linear systems, validate it on non-PDE matrices (e.g., SuiteSparse).
- Does SymMaP require instance-specific training? If features are different (Darcy Flow Problem, Second-order Elliptic Partial Differential Equation, Biharmonic Equation, Poisson Equation, ...), what architectural changes ensure cross-instance generalization?

**Ethical Concerns:**

["NO or VERY MINOR ethics concerns only"]

**Final Justification:**

Thank you for your detailed response. Your reply has addressed the questions I raised and clarified my understanding of your work.

My initial concerns have been resolved, and I do not have additional issues after the rebuttal and discussion.

Therefore, I maintain my original positive score of 4.

**Limitations:**

Yes

**Quality:**

3

**Strengths And Weaknesses:**

Strengths:
- The learned symbolic expressions are concise, interpretable, and incur almost no computational overhead, making them suitable for integration into existing solver libraries.
- SymMaP demonstrates strong generalization across various PDE types, matrix sizes, and preconditioning methods, outperforming both fixed-parameter and neural network baselines in terms of both accuracy and speed.
-  The symbolic expressions can be compiled and directly embedded into CPU-based solvers (e.g., PETSc), addressing a key limitation of ML-based approaches that often require GPU acceleration.
- The approach enhances transparency in scientific computing by providing explicit formulas that can be analyzed for theoretical soundness and safety.

Weaknesses:
- While the paper compares with basic MLPs and some recent learning-based methods, it does not benchmark against the latest large-scale neural symbolic regression models or hybrid approaches that might offer stronger baselines.
- Evaluated exclusively on PDE-derived systems; untested on broader linear algebra benchmarks (e.g., network problems, optimization tasks).

---

> ### Author Rebuttal · Authors · 2025-07-30
>
> **Dear Reviewer,**
>
> We sincerely thank you for your thoughtful review. Your insightful feedback is invaluable, and we are grateful for the opportunity to clarify our method's capabilities, address your concerns, and discuss its future potential. Your comments have helped us identify key areas for improvement and further exploration.
>
> We provide detailed responses to your points below.
>
> ---
>
> ### **On Comparison with Advanced Symbolic** **Regression** **Baselines (Weakness 1)**
>
> We thank the reviewer for this valuable suggestion. Our paper's primary contribution is the novel framework that applies symbolic discovery to the problem of matrix preconditioner parameterization, rather than proposing a new symbolic regression algorithm itself.
>
> Our main goal in the comparison with a standard MLP was to validate a core premise: that concise symbolic expressions possess sufficient expressive power to accurately model the relationship between problem features and optimal parameters in this domain. Our results confirm this viability.
>
> We fully agree that the symbolic learning module within SymMaP is modular and can be replaced. The prefix notation + RNN approach we used is a well-established and effective paradigm. We believe our experiments demonstrate its strong performance for this task. We are excited by the prospect that future breakthroughs in neural symbolic regression can be readily integrated into the SymMaP framework, potentially unlocking even greater performance. We will clarify this aspect in the final version.
>
> ### **On Generalization, Evaluation Scope, and Architectural Design (Weakness2, Questions 2 & 3)**
>
> We thank the reviewer for these insightful questions regarding the evaluation scope and the mechanism for cross-instance generalization. We are happy to elaborate on our methodology and future vision.
>
> 1. On the Choice of Problem Domain and Benchmarks (PDEs & PETSc):
>
> Our choice to focus on PDE-derived systems and use PETSc as the solver environment was a deliberate one, intended to ground our work in the context of recent, high-impact research. Our experimental design is closely aligned with that of prominent papers in the field, such as [1] (NeurKItt, NeurIPS 2024) and [2](ICLR 2024).
>
> - **PETSc** is a highly-optimized, professional library for solving linear systems, particularly those arising from differential equations. It is the foundation for industrial software like FEniCS and deal.II, making it a relevant and rigorous benchmark environment. Specialized solvers like Gurobi are designed for different problem classes (e.g., linear programming), while low-level libraries like MKL do not provide the same high-level preconditioning framework.
> - **Differential equations** represent a vast and critical class of problems in science and engineering. Many optimization problems, such as the **Markov chain** **(Boltzmann Motion)** example discussed on **Page 18 (Line 677)** and evaluated on **Page 26**, are also formulated or solved using differential equations.
> - In principle, SymMaP is a symbolic discovery framework that is agnostic to the problem's origin (e.g., PDE discretization, linear programming). However, as you correctly imply, its practical application is currently bounded by the performance limitations of symbolic regression, which struggles with an excessively high number of input features (e.g., >1000). This makes SymMaP, in its current form, less suitable for problems characterized by a vast number of discrete and unstructured parameters, such as large-scale integer programming problems in optimization tasks.
>
> 2. On the Mechanism for Cross-Instance Generalization:
>
> As noted in the paper (Page 5, Line 157), SymMaP is a data-driven framework that learns a mapping from a problem's input features to its optimal preconditioner parameters. For this learning process to succeed, the features must be semantically consistent across the dataset. Therefore, a single learned expression is specific to a class of problems that share a common parametric description (e.g., all second-order elliptic PDEs, characterized by their coefficients).
>
> This requirement explains why we cannot directly train or test on a benchmark like SuiteSparse, which is a collection of individual, structurally diverse matrices without a shared, low-dimensional feature space. The architecture of SymMaP itself is general; it does not require changes between problem types. However, it does require a consistent feature representation for each training task.
>
> 3. On the Path to Broader Generalization (Future Work):
>
> You have raised an excellent point about extending SymMaP to handle a wider variety of linear systems and enable joint training across different problem types. Addressing this is a core part of our future research agenda. Our planned approach is to move from high-level equation parameters to more fundamental matrix properties as input features. We plan to learn the symbolic relationship between low-cost, obtainable matrix features (e.g., matrix norms, trace, sparsity patterns) and the optimal preconditioner parameters. This would create a universal feature space, breaking the dependency on problem-specific coefficients and enabling generalization across a much broader range of linear algebra problems. While directly using all matrix elements is currently infeasible, we are actively exploring methods to automatically select the most informative matrix features. This will make SymMaP a truly universal tool for a much wider class of linear systems.
>
> **References:**
>
> [1] Neural krylov iteration for accelerating linear system solving, NIPS 2024.
>
> [2] Accelerating data generation for neural operators via krylov subspace recycling, ICLR 2024
>
> ### **On Revisions to Figure 1 (Question 1)**
>
> Thank you for your sharp eye and for pointing this out. We have re-examined Figure 1 and confirmed that the misalignment of the "78%" labels was indeed due to a decimal rounding issue in the plotting script. We sincerely apologize for this oversight. In the final version of the paper, we will correct this and ensure all fonts are standardized to Times New Roman for consistency and professionalism.
>
> ----
> ## **Thanks Again**
>
> We sincerely thank you again for your constructive and detailed feedback, which has helped us identify clear pathways to improve our paper. Should you have any further questions or require additional discussion, please don't hesitate to reach out. If we have adequately addressed your concerns, we would be grateful for your consideration in adjusting your evaluation score accordingly.

---

> > ### Comment · Reviewer_azLf · 2025-08-05
> >
> > Thank you for your detailed response. Your reply has indeed addressed the questions I raised and clarified my understanding of your work.

---

### Official Review · Reviewer_C25p · 2025-07-01

**Clarity:** 2
**Significance:** 2
**Originality:** 4
**Rating:** 4
**Confidence:** 3

**Summary:**

This paper introduces SymMaP, a symbolic discovery framework for automatically learning interpretable, efficient symbolic expressions to predict matrix preconditioning parameters for linear solvers. Traditional fixed-parameter methods lack adaptability, while machine-learning-based approaches suffer from high inference costs and poor interpretability. SymMaP combines the benefits of both worlds by using a neural network to explore a symbolic search space and generate compact mathematical formulas describing optimal preconditioning parameters.

**Questions:**

1. Could it be extended for designing mechanisms for dynamic or adaptive parameter updating, which are often necessary in time-dependent or evolving problems?
2. Please kindly refer to some questions in the weaknesses above.

**Ethical Concerns:**

["NO or VERY MINOR ethics concerns only"]

**Final Justification:**

My final justification for the recommended score is still 4 (borderline accept). My questions have been answered, and I appreciate the authors' detailed responses.

**Limitations:**

yes

**Paper Formatting Concerns:**

I did not find major formatting issues.

**Quality:**

3

**Strengths And Weaknesses:**

Strengths
- This paper aims to find a good tradeoff between interpretability and performance. Because the output is an interpretable symbolic model, users could understand and analyze the learned preconditioning strategies. Experimental results show consistent improvements over traditional fixed-parameter baselines on second-order elliptic PDE problems.
- The learned expressions can be compiled into a shared library and integrated into existing solver frameworks with minimal engineering effort. Meanwhile, this method is computationally efficient at inference time.

Weaknesses:
- The paper claims that the framework is not strictly tied to SOR and could potentially generalize to other continuous-parameter preconditioners, but I am still worry about its generalization ability. I am wondering how well the learned symbolic expressions perform on linear systems from significantly different domains, like other type of PDE problems.
- In real industrial scenarios, obtaining large-scale training data through extensive grid search can be computationally expensive or infeasible for new or large problems, which may limit the method’s practicality.
- It remains unclear whether the method can be effectively applied to real-world problems such as linear programming, where the structure of the linear systems can be very different from PDE discretizations.
- While the method produces symbolic expressions that are claimed to be interpretable, it is not always clear how these formulas correspond to the physical properties of the original problem. This may limit their practical usefulness in terms of interpretability. In some industrial settings, interpretability might be less prioritized compared to achieving faster convergence. Practitioners may run multiple preconditioners in parallel and select the fastest one, accepting the extra resource use to reduce overall end-to-end time. Therefore, there is a possibility that focusing on interpretability could trade off some potential performance benefits offered by less interpretable, purely empirical approaches.

---

> ### Author Rebuttal · Authors · 2025-07-30
>
> **Dear Reviewer,**
>
> We sincerely thank you for your thoughtful review and for recognizing the "excellent" originality of our work. Your insightful feedback is invaluable, and we are grateful for the opportunity to clarify our method's capabilities, address your concerns, and discuss its future potential. Your comments have helped us identify key areas for improvement and further exploration.
>
> We offer the following points in response to your questions and weaknesses.
>
> ---
>
> ### **On Generalization and Applicability to Diverse Problem Domains (Weakness 1 & 3)**
>
> As noted in the paper (Page 5, Line 157), the current implementation of SymMaP takes the parameters of the governing PDE as input features. This approach requires that different problems can be described by a consistent set of parameters. Therefore, if different types of PDEs cannot be modeled using a unified set of parameters, our current model cannot be trained on them simultaneously. This is a limitation of the current feature engineering, not a fundamental constraint of the SymMaP framework itself.
>
> In principle, SymMaP is a symbolic discovery framework that is agnostic to the problem's origin (e.g., PDE discretization, linear programming). However, as you correctly imply, its practical application is currently bounded by the performance limitations of symbolic regression, which struggles with an excessively high number of input features (e.g., >1000). This makes SymMaP, in its current form, less suitable for problems characterized by a vast number of discrete and unstructured parameters, such as those found in large-scale integer programming.
>
> This is a very important point, and your question highlights a crucial direction for our future research. We plan to address this by shifting our focus from high-level equation parameters to more fundamental matrix properties. Our next step is to use easily computable matrix features (e.g., matrix norms, trace) and perhaps a curated subset of key matrix elements as inputs to the symbolic learning process. While directly using all matrix elements is currently infeasible, we are actively exploring methods to automatically select the most informative matrix features. This will make SymMaP a truly universal tool for a much wider class of linear systems.
>
> ### **On Data Efficiency and Adaptability to Dynamic Problems (Weakness 2 & Question 1)**
>
> You are correct that SymMaP is a data-driven method. However, a key advantage of symbolic learning is its remarkable data efficiency. As shown in **Table 16 (Appendix, Page 29)**, SymMaP can achieve strong performance with a very small dataset. For instance, learning an effective AMG preconditioner for the Darcy Flow problem required **only 100 data points**. For the main experiments in the paper, we conservatively used a larger dataset (n=1000) to ensure the utmost rigor and stability of our results, but this is not the minimum requirement.
>
> Your suggestion to extend SymMaP to dynamic or adaptive scenarios is an excellent one and aligns perfectly with our vision for future work. We are planning to develop an online, adaptive learning extension of SymMaP. The idea is to first learn a foundational symbolic expression from a small, offline-generated dataset. Then, in a live operational setting, this expression would be continuously fine-tuned using reinforcement learning based on the performance on new, incoming problem instances. This approach would achieve two goals:
>
> 1. It would further minimize the reliance on extensive initial training data.
> 2. It would allow the preconditioning strategy to adapt to evolving problems, making it highly suitable for time-dependent and other dynamic industrial scenarios.
>
> ### **On the Balance of Interpretability and Performance (Weakness 4)**
>
> We appreciate the reviewer's nuanced perspective on the trade-off between interpretability and performance. We agree that in many industrial applications, computational efficiency is the highest priority. Our goal with SymMaP is to demonstrate that these two objectives are not necessarily mutually exclusive.
>
> The primary advantage of SymMaP is not to propose a new standalone preconditioner, but rather to serve as a **versatile framework that can be** **embedded** **within any parameterized preconditioner to enhance its performance.** As we demonstrate in our **related work comparison (Appendix E.6, Pages 25-27)**, many learning-based methods focus on optimizing one specific algorithm (e.g., a particular type of AMG [1] or ICC [2]). If that underlying algorithm is a poor fit for a given problem, the learning-based enhancement has a limited ceiling. SymMaP, by contrast, offers broader applicability.
>
> More importantly, SymMaP is not in opposition to other learning-based methods; it is complementary. Our results in **Table 12 and 13 (Appendix, Page 26)** show this clearly.
>
> - The **best overall performance** across our benchmarks was achieved by combining SymMaP with another learning-based method ("NeurKItt [3] + SymMaP"). This shows that SymMaP can be a critical component in a state-of-the-art pipeline.
> - When evaluating **standalone algorithms**, SymMaP consistently delivered the best performance, outperforming other individual methods.
>
> These results strongly indicate that SymMaP provides a significant performance benefit. Interpretability is an additional, powerful advantage. It allows researchers to analyze the learned formulas (some of which are remarkably simple, as shown on Page 28) to gain deeper insights into the preconditioning process itself. In fact, we are actively pursuing this, with the goal of having SymMaP's discoveries inform the theoretical design of new and improved preconditioning algorithms.
>
> **References:**
>
> [1] Learning to optimize multigrid pde solvers, ICML 2019.
>
> [2] Neural incomplete factorization: learning preconditioners for the conjugate gradient method, TMLR 2024.
>
> [3] Neural krylov iteration for accelerating linear system solving, NIPS 2024.
>
> ----
> ## **Thanks Again**
>
> We sincerely thank you again for your constructive and detailed feedback, which has helped us identify clear pathways to improve our paper. Should you have any further questions or require additional discussion, please don't hesitate to reach out. If we have adequately addressed your concerns, we would be grateful for your consideration in adjusting your evaluation score accordingly.

---

> > ### Comment · Reviewer_C25p · 2025-08-04
> >
> > Thank you to the authors for the detailed responses, explanations of limitations, and future steps! They have answered all my questions.

---

### Official Review · Reviewer_BuLC · 2025-07-05

**Clarity:** 3
**Significance:** 3
**Originality:** 3
**Rating:** 4
**Confidence:** 2

**Summary:**

This paper presents a symbolic-based machine learning method for identifying optimal parameters for matrix preconditioning. The method is described in detail and offered as an end-to-end solution. Experimental results demonstrate that it can identify parameter values close to the optimum, thereby enhancing the adaptability of linear system solvers across different tasks.

**Questions:**

Please find my questions in the "Strengths and Weaknesses" section. Thanks.

**Ethical Concerns:**

["NO or VERY MINOR ethics concerns only"]

**Final Justification:**

I remain positive about this paper after the response period.

**Limitations:**

The authors briefly discuss the limitations of their approach, but there are several important aspects that warrant further attention—such as the manual effort currently required to integrate the symbolic method's output into solvers, and the limited range of supported solvers. Expanding the limitations section to address these points in more depth would significantly strengthen the paper.

**Paper Formatting Concerns:**

No.

**Quality:**

3

**Strengths And Weaknesses:**

Strengths:

1. The paper addresses a fundamental problem in optimization and machine learning. While the topic may not be currently trending, it holds significant value in the domains of High-Performance Computing (HPC) and AI for Science.

2. The proposed symbolic approach offers a degree of explainability compared to traditional black-box ML methods for searching optimal parameters in matrix preconditioning. The method is well implemented, featuring a complete pipeline: data generation, parameter search using reinforcement learning (RL), and a thoughtful state representation via RNN. The output is integrated into a linear system solver to demonstrate practical benefits.

3. The paper includes sufficient background material, making it accessible even to non-expert readers. The writing is clear and approachable, which is commendable.


Weaknesses:

1. I am somewhat concerned about the limited description of how the symbolic method integrates into real-world linear system solvers. The paper currently lacks detail in this area. It would be highly beneficial to include a more thorough explanation—ideally, demonstrating integration with at least two widely used solvers. This would help the audience better understand the practical applicability and impact of the proposed method.

2. The evaluation scope is relatively narrow, a limitation that the authors acknowledge in the conclusion. Given that the symbolic method should theoretically generalize better to different matrix preconditioning setups—due to its reduced reliance on training data—it's disappointing that this potential is only discussed rather than thoroughly evaluated. A more comprehensive evaluation on diverse preconditioning scenarios would significantly enhance the paper's contribution, especially in terms of its claimed generalizability and explainability.

3. Several design choices, such as the use of adaptive grid search and RNNs, are mentioned but not sufficiently justified or compared against plausible alternatives. It would strengthen the paper to include discussion and experimental results demonstrating why these choices are effective or superior to other options.

4. Did you consider including conventional hyperparameter search methods (e.g., Bayesian optimization, random search, or grid search) in your evaluation? These methods are common baselines in this area and should be included to contextualize the performance of your symbolic-RL approach.

---

> ### Author Rebuttal · Authors · 2025-07-30
>
> **Dear Reviewer,**
>
> We are sincerely grateful for your thorough review and insightful feedback on our paper. Your comments are invaluable, and we appreciate the opportunity to clarify our contributions and address your concerns. Below, we address each of your points in detail.
>
> ----
>
> ### **On Practical Integration with Real-World Linear Solvers  (Weakness 1)**
>
> The process of integrating our algorithm is a very simple, two-step procedure:
>
> 1. Parameter Computation via the Symbolic Expression:
>
> Once SymMaP discovers a symbolic formula, applying it to a new problem is computationally trivial. The problem's specific features are simply substituted into the learned expression to calculate the recommended preconditioner parameter. This can be summarized with the following pseudo-code: (Python)
>
> ```Plain
> def calculate_preconditioner_param(problem_features, learned_expression):"""
>     Calculates the optimal preconditioner parameter using the learned symbolic expression.
>
>     Args:
>         problem_features (dict): A dictionary of problem features, e.g., {'x1': 0.5, 'x2': 1.2}.
>         learned_expression (str): The formula discovered by SymMaP, e.g., "1.8" or "0.5 + 0.9 * x1".
>
>     Returns:
>         float: The computed preconditioner parameter.
>     """
>     param_value = safe_evaluate(learned_expression, variables=problem_features)
>     return param_value
> ```
>
> 2. Seamless Integration with Standard Solver Libraries:
>
> The computed parameter can be directly passed to mainstream solver libraries with no modification to the solvers themselves. We demonstrate this with two widely-used examples, as you suggested:
>
> - **Integration with PETSc (via command-line):** PETSc is a standard toolkit in HPC. A user can pass the computed relaxation factor `omega` directly to a PETSc-based application using the `-pc_sor_omega` flag: (Bash)
>
> ```Plain
> # The value '1.8' is calculated by our method beforehand
> ./my_petsc_solver -ksp_type gmres -pc_type sor -pc_sor_omega 1.8
> ```
>
> - **Integration with PyAMG (via Python** **API****):** PyAMG is a popular Python library for Algebraic Multigrid methods. The computed strength threshold `theta` can be set directly in the solver configuration: (Python)
>
> ```Plain
> import pyamg
> # The value '0.25' is calculated by our method
> optimal_theta = 0.25
> # Pass the parameter directly into the PyAMG solver configuration
> strength_config = ('symmetric', {'theta': optimal_theta})
> ml = pyamg.smoothed_aggregation_solver(A, strength=strength_config)
> ```
>
> The computational overhead of this integration is limited to a single evaluation of the symbolic expression, which is negligible compared to the cost of the iterative linear solve. The complete code for our experiments is already available in the supplementary material.
>
> **Commitment:** We agree that this is an important practical detail. In the final version of the paper, we will add a dedicated section to thoroughly describe this integration process. Furthermore, we will package our code into a user-friendly C/Python function to facilitate its adoption and potential integration into official preconditioning libraries.
>
> ### **On Broadening the Evaluation Scope and Generalizability (Weakness 2)**
>
> Our initial focus was on SOR, SSOR, and AMG because they exhibited a non-monotonic relationship between their parameters and performance metrics (e.g., condition number or solution time). This presents a non-trivial optimization problem where a clear optimal parameter exists, making them ideal candidates for showcasing SymMaP's core capability.
>
> We acknowledge that other important preconditioners, such as ICC and ILU, present a different challenge. For these, increasing a parameter like the fill-in level typically decreases the condition number but monotonically increases the computational cost of the factorization. A real-world application requires balancing this trade-off, which involves defining a custom, often heuristic, objective function. To avoid introducing ambiguity with such custom metrics in our initial submission, we prioritized the more clear-cut optimization cases.
>
> To further address your concern and demonstrate the broader applicability of SymMaP, we have conducted a new experiment on **ICC** **preconditioning**. We defined a hybrid metric: `Objective = 0.03 * Condition_Number + Time(s)` to find the optimal fill-in level. The experiment was run on the symmetric second-order elliptic PDE (matrix size 40,000), with other settings consistent with paper.
>
> | Method          | None   | Default | Optimal Constant | SymMaP |
> | --------------- | ------ | ------- | ---------------- | ------ |
> | Objective Value | 227.66 | 24.01   | 22.74            | 22.27  |
>
> These results confirm that SymMaP can effectively optimize parameters for different preconditioners, even under custom, multi-objective metrics.
>
> **Commitment:** In the final version, we will incorporate the analysis above and expand our experiments to cover more preconditioners (including ILU) with custom objective functions. We will also release the corresponding code to demonstrate SymMaP's flexibility. We believe this will substantially strengthen the paper's claims on generalizability.
>
> ### **On Justifying Design Choices and Comparing with Baselines (Weakness 3 & 4)**
>
> Our primary focus was on demonstrating the efficacy of the symbolic learning framework itself, for which our chosen methods proved effective. The use of an RNN is a natural choice for processing sequential prefix notation data. However, the reviewer raises an excellent point that the data generation step could be performed by other established search methods.
>
> To address this, we conducted a comparative experiment on the data generation step for the 2nd-order elliptic PDE with SOR (matrix size 40,000, dataset size 1,000). We compared our adaptive grid search against standard grid search, random search, and Bayesian optimization.
>
> | Data Generation Method       | Avg. Generation Time | Mean Absolute Error (vs. our method) | Final Avg. Solution Time (s) |
> | ---------------------------- | -------------------- | ------------------------------------ | ---------------------------- |
> | Standard Grid Search         | ~100h                | 7.80E-04                             | 16                           |
> | Random Search (2000 samples) | ~100h                | 1.74E-03                             | 16                           |
> | Bayesian Optimization        | ~11h                 | 9.80E-04                             | 15.9                         |
> | Our Adaptive Search          | ~25h                 | 0                                    | 15.9                         |
>
> This analysis leads to three key conclusions:
>
> 1. **Consistent Optimal Parameters:** All search methods, when run to sufficient precision, converge to nearly identical optimal parameters (MAE < 1.74e-3).
> 2. **Consistent Symbolic Discovery:** Since the training datasets are virtually identical, SymMaP discovers the same symbolic expressions (differing only by minute constant values), resulting in nearly identical final performance (a 0.1s difference is within the margin of error).
> 3. **Efficiency of Alternatives:** As you astutely pointed out, other methods like Bayesian optimization are significantly more efficient for generating the training data.
>
> This is a valuable insight.
>
> **Commitment:** In the final version, we will update the paper to include this comparative analysis, explicitly justifying our design choices and contextualizing them with these standard baselines. We will also state that our framework is agnostic to the search method used for data generation and will provide multiple search strategies in our publicly released code to offer users more flexibility.
>
> ----
> ## **Thanks Again**
>
> We sincerely thank you again for your constructive and detailed feedback, which has helped us identify clear pathways to improve our paper. Should you have any further questions or require additional discussion, please don't hesitate to reach out. If we have adequately addressed your concerns, we would be grateful for your consideration in adjusting your evaluation score accordingly.

---

### Decision · Program_Chairs · 2025-09-17

**Decision:**

Accept (poster)

**Comment:**

This paper considers the important task of matrix preconditioning, a crucial step in linear system solving. While matrix preconditioners are frequently designed in an ad-hoc manner, the paper proposes to symbolically learn preconditioners to automate this process. Experimental results support the proposed method over prior strategies.

Reviewers and I find the problem relevant, and the proposed approach original and of interest to the community. I'm happy to recommend acceptance.